# CO anthropogenic emissions in Europe from 2011 to 2021: insights from Measurement of Pollution in the Troposphere (MOPITT) satellite data

**Audrey Fortems-Cheiney**[1,a], **Gregoire Broquet**[1], **Elise Potier**[1,b], **Robin Plauchu**[1], **Antoine Berchet**[1], **Isabelle Pison**[1], **Hugo Denier van der Gon**[2], **and Stijn Dellaert**[2]

[1]Laboratoire des Sciences du Climat et de l'Environnement, CEA-CNRS-UVSQ, Gif-sur-Yvette, France
[2]Department of Climate, Air and Sustainability, TNO, P.O. Box 80015, 3508 TA Utrecht, the Netherlands
[a]now at: Science Partners, Quai de Jemmapes, 75010 Paris, France
[b]now at: Laboratoire Inter-universitaire des Sciences Atmosphériques, Université Paris Est Créteil and Université Paris Cité, CNRS, LISA, 94010 Créteil, France

**Correspondence:** Audrey Fortems-Cheiney (audrey.fortems@science-partners.com)

**Abstract.** We have used the variational inversion drivers of the recent Community Inversion Framework (CIF), coupled to a European configuration of the CHIMERE regional chemistry transport model and its adjoint to derive carbon monoxide (CO) emissions from Measurement of Pollution in the Troposphere (MOPITT) TIR-NIR (thermal-infrared near-infrared) observations, for a period of over 10 years from 2011 to 2021. The analysis of the inversion results reveals the challenges associated with the inversion of CO emissions at the regional scale over Europe. Annual budgets of national emissions have decreased by about 1 %–11 % over the decade and across Europe. These decreases are mainly due to negative corrections during autumn and winter. The posterior CO emissions follow a decreasing trend over the European Union and United Kingdom area of about $-2.2\,\%\,\mathrm{yr}^{-1}$, slightly lower than in the prior emissions. The assimilation of the MOPITT observation in the inversions indeed attenuates the decreasing trend of the CO emissions in the TNO inventory over areas benefiting from the highest number of MOPITT super-observations (particularly over Italy and over the Balkans), and particularly in autumn and winter. The small corrections of the CO emissions at national scales by the inversion can be attributed, first, to the general consistency between the TNO-GHGco-v3 inventory and the satellite data. Analysis of specific patterns such as the impact of the Covid-19 crisis reveals that it can also be seen as a lack of observation constraints to adjust the prior estimate of the emissions. The large errors associated with the observations in our inversion framework and the lack of data over large parts of Europe are sources of limitation on the observational constraint. Emission hotspots generate a relatively strong local signal, which is much better caught and exploited by the inversions than the larger-scale signals, despite the moderate spatial resolution of the MOPITT data. This is why the corrections of these hotspot emissions are stronger and more convincing than the corrections of the national- and continental-scale emissions. Accurate monitoring of the CO national anthropogenic emissions may thus require modelling and inversion systems at spatial resolutions finer than those used here as well as satellite images at high spatial resolution. The CO data of the TROPOMI instrument on board the Sentinel-5P mission should be well suited for such a perspective.

# 1 Introduction

Carbon monoxide (CO) is an air pollutant and a greenhouse gas, mainly emitted by anthropogenic activities and impacting both air quality and climate change. It plays a major role in atmospheric chemistry as a key component of the methane ($CH_4$) oxidation chain with formaldehyde (HCHO), ozone ($O_3$), and carbon dioxide ($CO_2$). Through chemical interactions with hydroxyl radical (OH), CO (i) influences concentrations of $CH_4$ and non-methane volatile organic compounds (NMVOCs), (ii) affects the self-cleaning or oxidation capacity of the atmosphere (Lelieveld et al., 2016), and (iii) leads to the chemical production of air pollutants and/or greenhouse gases such as tropospheric $O_3$ and $CO_2$. In this context, there is a need for accurate mapping or monitoring of the CO surface emissions.

CO emissions estimated by bottom-up (BU) inventories, based on statistical and economic data and relying on emission factors per activity type, suffer from relatively large uncertainties. For example, at the national and annual scales, these uncertainties range from 20 %–60 % to 50 %–200 %, depending on the sectors in the European Monitoring and Evaluation Programme (EMEP) inventory (Kuenen and Dore, 2019). Complementary to BU inventories, atmospheric CO concentration data, such as those observed from satellite observations, can be used to derive estimates of the CO fluxes based on atmospheric transport inverse modelling techniques (Rayner et al., 2019). Over the last 2 decades, the space-borne Measurement of Pollution in the Troposphere (MOPITT; Drummond et al., 1996; Deeter et al., 2003), the Atmospheric Infrared Sounder (AIRS; Aumann et al., 2003; McMillan et al., 2005), the Tropospheric Emissions Spectrometer (TES; Beer, 2006), and the Interféromètre Atmosphérique de Sondage dans l'Infrarouge (IASI; Clerbaux et al., 2009) have revolutionized our ability to map CO concentrations and to understand the trends and spatiotemporal variability of its concentrations and emissions (Arellano et al., 2006; Chevallier et al., 2009; Jones et al., 2009; Kopacz et al., 2010; Jiang et al., 2011; Fortems-Cheiney et al., 2011; Hooghiemstra et al., 2012; Miyazaki et al., 2015; George et al., 2015; Yin et al., 2015; Jiang et al., 2017; Zheng et al., 2018; Buchholz et al., 2021; Gaubert et al., 2023). However, the potential of satellite data to inform about CO emissions has mainly been explored at the global scale, with emission estimates corresponding to large regions. Today, scientific and societal issues require an up-to-date quantification of pollutant emissions at a higher spatial resolution targeting national estimates. This currently requires the use of regional-scale inversion systems (Fortems-Cheiney et al., 2021).

However, although these systems are suited to reactive species, they have hardly been used to quantify emissions of pollutants such as CO. In the past decade, CO regional-scale inversions based on the MOPITT data covered the CO emissions in North America (Jiang et al., 2015) and East Asia (Qu et al., 2022). To our knowledge, there have only been a few studies covering the European CO emissions based on satellite observations (Konovalov et al., 2016; Fortems-Cheiney et al., 2021), this continent being more challenging for regional-scale inversions of the CO anthropogenic emissions owing to a weaker CO signal (Konovalov et al., 2016). Konovalov et al. (2016) estimated CO European emissions from the IASI thermal-infrared (TIR) satellite measurements over Europe but pointed out the low sensitivity of the corresponding CO total columns to anthropogenic CO emissions. Deeter et al. (2013) showed that the sensitivity of the total columns to CO emissions in the lower troposphere – where the regional signal from CO regional anthropogenic emissions above the large-scale and highly mixed CO background is largest – should be significantly greater for retrievals exploiting simultaneous TIR and near-infrared (NIR) measurements than for retrievals based on either spectral region alone. Fortems-Cheiney et al. (2021) performed regional inversions using MOPITT TIR-NIR satellite observations over Europe to illustrate the behaviour of the variational atmospheric inversion system PYVAR-CHIMERE, but only over a short temporal window of 7 d. The ability of regional inverse systems to quantify CO budgets at the national and monthly to annual scales in Europe from the MOPITT TIR-NIR satellite observations has not been assessed yet.

The objective of this work is therefore to carry out a long-term regional inversion for Europe using these observations. We estimate CO emissions from the MOPITT TIR-NIR observations for more than 10 years from January 2011 to November 2021. The analysis over the period 2011–2021 makes it possible to evaluate the strong trends indicated by the BU inventories over the decade and major inter-annual anomalies, in particular the expected reduction of emissions in 2020 due to the measures taken in response to the Covid-19 pandemic. For this objective, we have used the variational inversion drivers of the recent Community Inversion Framework (CIF; Berchet et al., 2021), which inherits the developments made for the regional assimilation of satellite data on gaseous species by Fortems-Cheiney et al. (2021). We also use a European configuration of the CHIMERE regional chemistry transport model (CTM) (Menut et al., 2013; Mailler et al., 2017) and of its adjoint (Fortems-Cheiney et al., 2021) driven by the CIF. The data and methods used in this study are described in Sect. 2. The results are described in Sect. 3.

# 2 Data and methods

## 2.1 Configuration of the CHIMERE CTM for the simulation of CO concentrations in Europe

The configuration of the atmospheric CHIMERE CTM for Europe is described in Table 1. CHIMERE is run over a $0.5° \times 0.5°$ regular horizontal grid and 17 vertical layers, from the surface to 200 hPa, with 8 layers within the first 2 km. The domain covers Europe (15.25° W–35.75° E,

31.75–74.25° N) and includes 101 (longitude) × 85 (latitude) grid cells. The ERA-Interim reanalyses – the only ones available at the beginning of this study – remain at the rather low horizontal resolution of 79 km compared to the forecast fields. Consequently, as a trade-off between the accuracy of large-scale meteorological fields and resolutions at finer resolution, CHIMERE is driven here by the European Centre for Medium-Range Weather Forecasts (ECMWF) operational meteorological forecast (Owens and Hewson, 2018), with a spatial resolution of 0.25°. The chemical scheme used in CHIMERE is MELCHIOR-2, with more than 100 reactions (CHIMERE, 2017), including the secondary production of CO through the oxidation and photolysis of hydrocarbons and its sink with OH.

Initial and boundary conditions for several key gaseous species responsible for the oxidation capacity of the lower atmosphere (e.g. CO, NO, $NO_2$, $O_3$, $H_2O_2$, or HCHO) were specified using monthly climatological data from the LMDz-INCA global model (Szopa et al., 2008).

CO emissions from fires, which account for about 2 % of the total European CO emissions (San-Miguel-Ayanz and Steinbrecher, 2019), are not taken into account in this study. CO biogenic emissions are assumed to be negligible and are not taken into account. In contrast to Fortems-Cheiney et al. (2021) using TNO-GHGco-v1, the prior estimate of CO anthropogenic emissions is derived from the recent TNO-GHGco-v3 gridded inventory for the period 2011–2018. The TNO-GHGco version is an update of the TNO inventory (Super et al., 2020; Denier van der Gon et al., 2021; Kuenen et al., 2022) based on EMEP/CEIP official country reporting for air pollutants. This inventory has been delivered with an extrapolation of the emissions for the year 2019 based on an in-sample approach (Super et al., 2020). We use this combination of products for the years 2011–2019. Our prior estimates of the emissions for 2020 and 2021 are set at the values for 2019. The horizontal resolution of the TNO-GHGco-v3 inventory is $6 \times 6 \, km^2$. The TNO-GHGco inventory combines emissions from area sources, set at the surface, and from point sources. Emissions from point sources, mainly from the energy production and industrial sectors, are distributed on the vertical model layers typically depending on the injection height provided in the TNO inventory, based on Bieser et al. (2011). The annual and national budgets from EMEP/CEIP are disaggregated in space based on proxies of the different sectors of activity (Kuenen et al., 2022). The temporal disaggregation is based on temporal profiles provided per Gridded Nomenclature for Reporting (GNFR) sector code with typical month-to-month, weekday-to-weekend, and diurnal (at a 1 h scale) variations. The TNO-GHGco-v3 inventory is aggregated at the $0.5° \times 0.5°$ horizontal resolution of the CHIMERE grid. The resulting prior anthropogenic CO emissions from 2011 to 2021 for the European Union + United Kingdom (EU-27+UK) area are illustrated in Fig. 1, and the resulting map of prior anthropogenic CO emissions is shown in Fig. 2a for January 2015. CO emissions are high over large cities and over industrial areas (e.g. over the Benelux, the Po Valley in Italy, north-western Germany, or southern Poland).

In addition to CO, the MELCHIOR-2 chemical scheme needs emissions from other species, such as NMVOCs or nitrogen oxides ($NO_x = NO + NO_2$). Anthropogenic $NO_x$ emissions are from the TNO-GHGco-v3 inventory, while NMVOC anthropogenic emissions are from the EMEP inventory (Vestreng et al., 2005). Biogenic $NO_x$ and NMVOC emissions, in particular emissions of isoprene and some other hydrocarbons from vegetation, are obtained from the Model of Emissions of Gases and Aerosols from Nature (MEGAN) (Guenther et al., 2006).

The resulting monthly mean volume mixing ratios between the surface and 900 hPa are illustrated in Fig. 3a for January 2015. The sensitivity of CO-simulated concentrations to CO emissions is evaluated by running a sensitivity test with European CO anthropogenic emissions set to zero: the simulated concentrations are illustrated in Fig. 4.

## 2.2 MOPITT satellite observations

CO inversions assimilate CO observations from the MOPITT retrieval product Version 8 (Deeter et al., 2019). MOPITT flies on board the NASA EOS-Terra satellite in a low sun-synchronous orbit that crosses the Equator at 10:30 and 22:30 local solar time (LST). The spatial resolution of its observations is about $22 \times 22 \, km^2$ at nadir. It has been operated nearly continuously since March 2000. MOPITT CO products are available in three variants: TIR only, NIR only, and the multispectral TIR-NIR product, all containing total columns and retrieved profiles (given on a 10-level grid from the surface to 100 hPa). Among the different MOPITTv8 products, we choose to work with the multispectral MOPITTv8-NIR-TIR one (also called MOPITT-v8J), as the sensitivity to CO in the lower troposphere should be significantly greater for retrievals exploiting simultaneous TIR and NIR measurements than for retrievals based on either spectral region alone (Worden et al., 2010; Deeter et al., 2013; Buchholz et al., 2017). In addition, it provides the highest number of data.

We choose to assimilate the MOPITT V8J surface product, derived as the mean volume mixing ratio between the surface and 900 hPa, as the surface level multispectral retrievals have greater sensitivity to CO near the surface and reduced sensitivity in the free troposphere (Jiang et al., 2015; Qu et al., 2022). Long-term trends of surface CO concentrations for 2001–2015 are very consistent between the "MOPITT lower profile" and World Data Center for Greenhouse Gases (WDCGG) sites (Jiang et al., 2017). The retrieval bias drift is also low at the surface level for V8 TIR-NIR products as compared to National Oceanic and Atmospheric Administration (NOAA) flask measurements (Deeter et al., 2019). Finally, the surface level of the V8 TIR-NIR products gives the low-

**Table 1.** Main characteristics of the European CHIMERE configuration used in this work.

| | |
|---|---|
| Domain | Europe (15.25° W–35.75° E, 31.75–74.25° N) |
| Horizontal resolution | 0.5° × 0.5° regular grid |
| Vertical resolution | 17 layers from the surface to 200 hPa |
| Meteorological fields | ECMWF operational meteorological forecast (Owens and Hewson, 2018) |
| Initial and boundary conditions | Climatological values from the LMDZ-INCA global model (Szopa et al., 2008) |
| Anthropogenic emissions | TNO-GHGco-v3 inventory (Super et al., 2020) |
| Biogenic emissions | MEGAN (Guenther et al., 2006) |

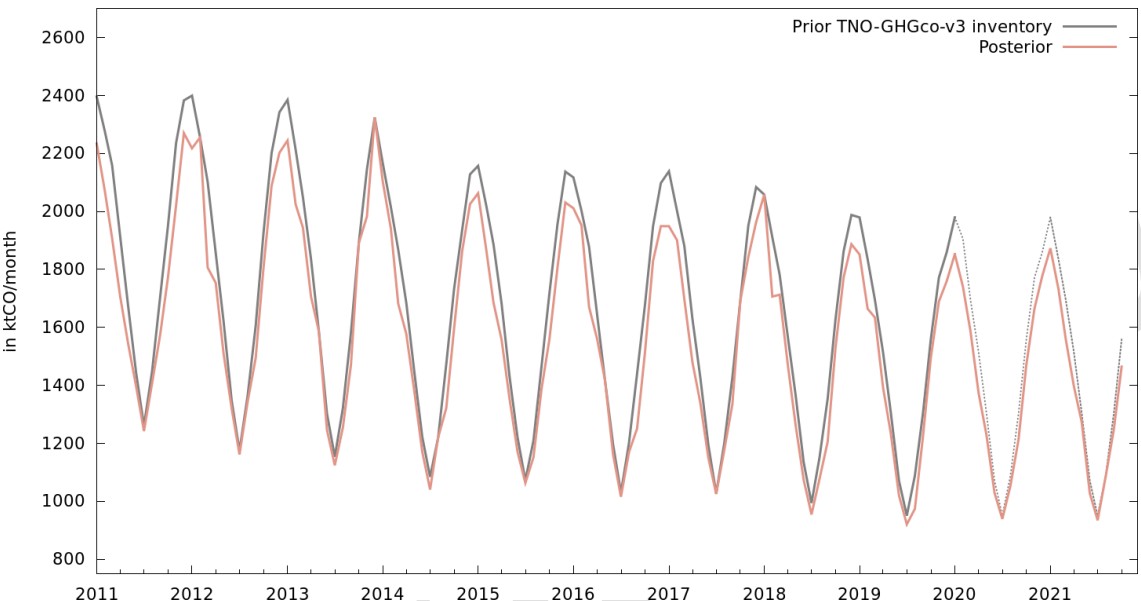

**Figure 1.** Estimates of the monthly budgets of CO for EU-27+UK from the TNO-GHGco-v3 inventory (solid light-grey line) and its extension to 2020–2021 (dashed light-grey line) as well as from the regional inversions (solid orange line) from January 2011 to December 2021.

est bias when compared to in situ data from NOAA aircraft validation sites (Deeter et al., 2019).

To make accurate comparisons between simulations and satellite observations, the averaging kernels (AKs) and the MOPITT prior profiles are applied to the simulated field so that the simulated concentrations exhibit the same degree of smoothing and a priori dependence as the MOPITT product (Deeter et al., 2013, 2019). Following the recommendations of Deeter (2018), the formula is applied: **TS1**

$$c_\mathrm{m} = x_\mathrm{a} + \mathbf{AK}\left(c_\mathrm{m}^\mathrm{o} - x_\mathrm{a}\right), \tag{1}$$

where $c_\mathrm{m}$ is the modelled column, $\mathbf{AK}$ **TS2** contains the averaging kernels – which are an indication of the vertical resolution of the measurements – provided in the form of a matrix, $x_\mathrm{a}$ is the prior profile derived from a model climatology and varies seasonally and geographically (Deeter et al., 2019), and $c_\mathrm{m}^\mathrm{o}$ is the vertical distribution of the original model partial columns interpolated to the pressure grid of the AKs.

In order to associate the super-observations with a real AK, the super-observations have been taken as the individual observation corresponding to the value of the median of

the MOPITT concentrations within the 0.5° × 0.5° grid cell of the CTM and within the CTM's physical time steps (about 5–10 min). The AKs and the uncertainty associated with this individual super-observation are then used to define the AK and the uncertainty for the "super-observation". In principle, the observation error associated with such a median value should be smaller than the error associated with an individual observation, but here we keep the error for the individual observation used to define the super-observation as a conservative estimate of the super-observation error. The super-observations therefore do not have a smaller error than the individual observations.

The resulting monthly means of the MOPITT super-observations and their simulated equivalents for CO average surface concentrations in January 2015 are respectively illustrated in Fig. 3b and c. The spatial patterns of the CO concentrations are very different if the MOPITT AK and prior profiles are applied (Fig. 3c) or not (Fig. 3a), particularly in central, eastern, and northern Europe. It shows that the MO-

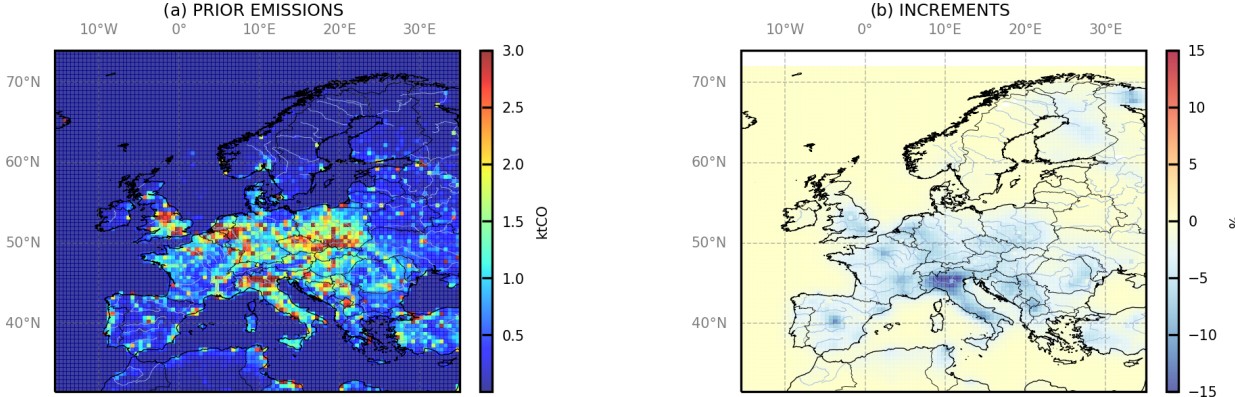

**Figure 2. (a)** Monthly CO emissions (kt CO) and **(b)** monthly mean relative increments to the TNO-GHGco-v3 inventory of CO anthropogenic emissions from the inversion in percentage, in January 2015, at the $0.5° \times 0.5°$ model resolution.

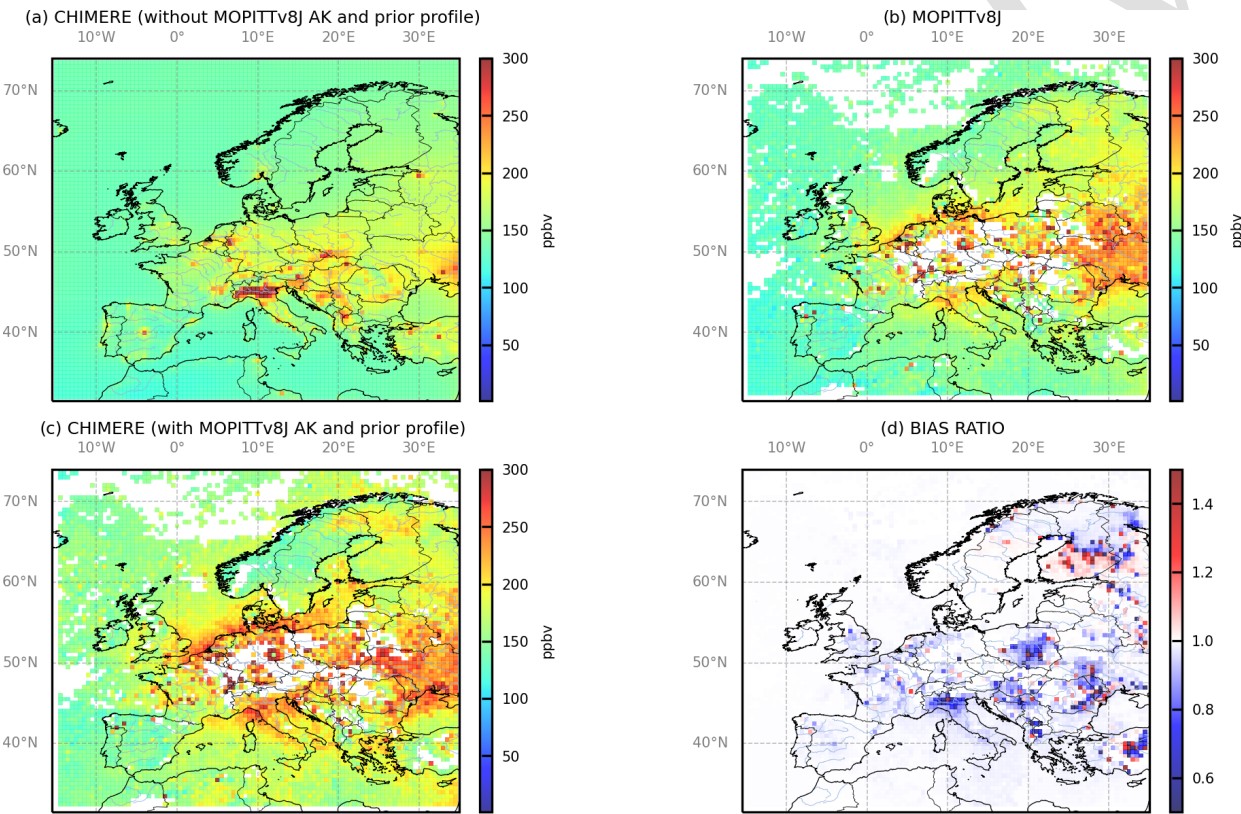

**Figure 3.** Averages of the CO concentrations between the surface and 900 hPa, **(a)** simulated by CHIMERE using the prior TNO-GHGco-v3 anthropogenic emission estimate without applying the MOPITT AK and prior profiles, **(b)** corresponding to the MOPITT surface super-observations in the CHIMERE grid, and **(c)** simulated by CHIMERE using the prior TNO-GHGco-v3 anthropogenic emission estimate applying the MOPITT AK and prior profiles (ppbv). **(d)** Ratios of the posterior and prior biases between monthly mean surface concentrations from CHIMERE and the MOPITT super-observations, at the $0.5° \times 0.5°$ grid cell resolution, in January 2015. All ratios lower than 1, in blue, demonstrate that posterior emission estimates improve the simulation compared to the prior ones.

PITT AK and prior profiles have a strong impact on the CO concentrations over these regions.

It is important to note that the potential of MOPITT to provide information can be strongly hampered by the cloud coverage in autumn and in winter, as illustrated in Fig. 3b with blanks for a large part of central Europe in January 2015. Generally, because of the cloud cover, the number of MOPITT super-observations is higher in the south of Europe than in central or northern Europe (Fig. 5a). The new MOPITT retrieval Version 9 product has a better observation coverage,

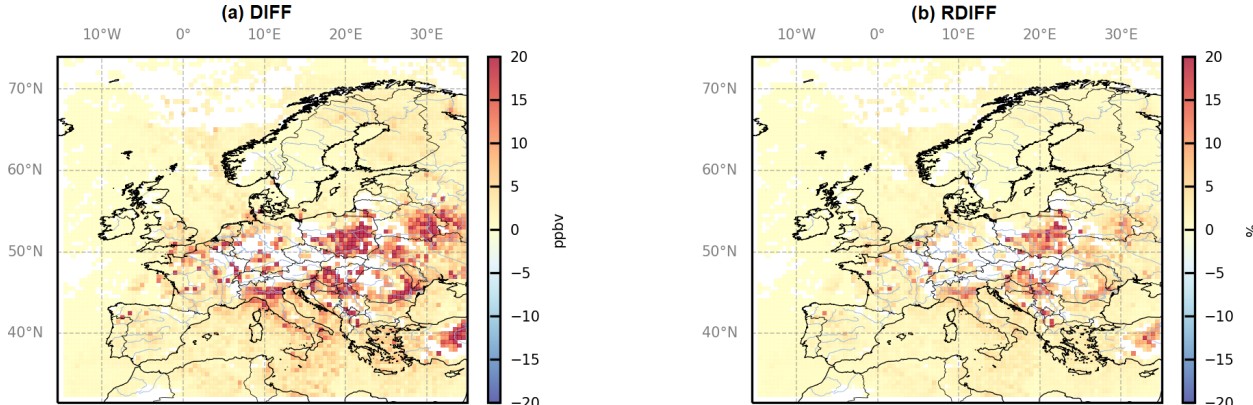

**Figure 4. (a)** Absolute differences (ppbv) and **(b)** relative differences (%) between the averages of CO concentrations simulated using the prior TNO-GHGco-v3 anthropogenic emission estimate and simulated with null CO emissions in January 2015.

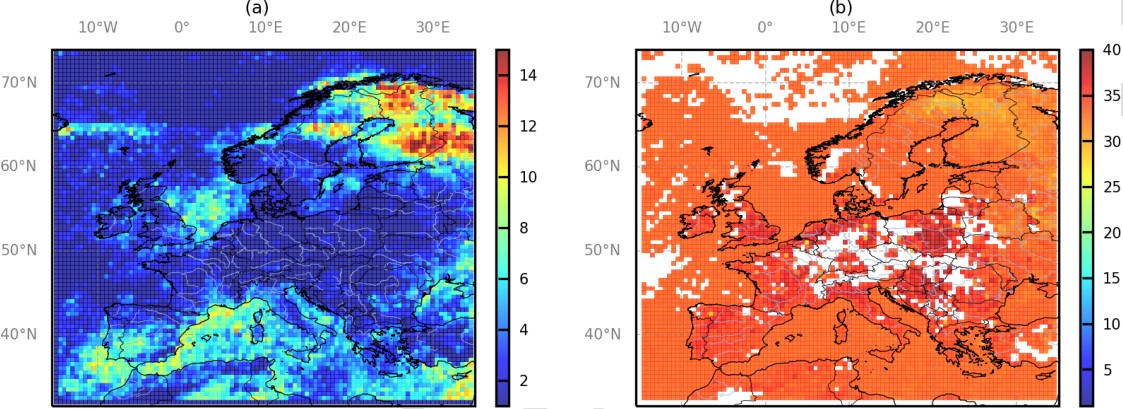

**Figure 5. (a)** Number of CO MOPITT super-observations and **(b)** averages of the errors associated with the CO MOPITT super-observations, in percentage, in January 2015.

with a number of daytime MOPITT retrievals over land increasing by 30 %–40 % relative to the Version 8 product due to improvements in the cloud detection algorithm (Deeter et al., 2022).

## 2.3 Variational inversion of CO anthropogenic emissions

The inversion of CO emissions consists in correcting the "prior" estimate of these emissions and of the model initial and/or boundary conditions to improve the fit between the simulated concentrations and the satellite CO data. The parameters of the variational inversions here closely follow the configuration of Fortems-Cheiney et al. (2021), which provides details on the principle and configuration for such inversions. The optimal ("posterior") estimate of the emissions in a statistical sense is found by iteratively minimizing the following cost function $J(x)$:

$$
\begin{aligned}
J(x) = &\frac{1}{2}\left(x - x^{\mathrm{b}}\right)^{T} \mathbf{B}^{-1}\left(x - x^{\mathrm{b}}\right) \\
&+ \frac{1}{2}(\mathcal{H}(x) - y)^{T}\mathbf{R}^{-1}(\mathcal{H}(x) - y),
\end{aligned} \tag{2}
$$

where $x$, $\mathcal{H}$, $y$, $\mathbf{B}$, and $\mathbf{R}$ are respectively the control vector, the observation operator, the observations, and the covariance matrices as detailed in the following paragraphs.

As a trade-off between computational resources and the relevance of our inversions with a moderate impact of the initial conditions on our 1-month CO simulation, series of independent 1-month inversion windows are run. We therefore do not account for the potential update of the concentrations during a previous 1-month window due to the inversions. Due to the relatively long lifetime of CO – i.e. a few weeks to 2 months (Prather, 1996) – compared to the size of the studied domain, we account for the CO lateral boundary conditions at the borders of the domain and for their uncertainties.

Therefore, the control vector $x$ contains the following.

– CO anthropogenic emissions at a 1 d temporal resolution, at a $0.5° \times 0.5°$ resolution and over the 17 vertical levels of CHIMERE, i.e. for a 1-month inversion, for each of the corresponding $(28 \text{ to } 31 \text{ d}) \times 101 \times 85 \times 17$ grid cells

- CO lateral boundary conditions at a 1 d temporal resolution, at a $0.5° \times 0.5°$ resolution, and over the 17 vertical levels of CHIMERE, i.e. for each of the corresponding $(28 \text{ to } 31\,\text{d}) \times 372 \times 17$ grid cells

- CO 3D initial conditions at 00:00 UTC on the first day of the month, at a $0.5° \times 0.5°$ resolution and over the 17 vertical levels of CHIMERE, i.e. for each of the corresponding $101 \times 85 \times 17$ grid cells.

It should be noted that the VOC emissions are fixed and not controlled here by the inversion. Nevertheless, the chemical production of CO by VOCs could be changed due to the correction of CO boundary conditions and fluxes through chemistry.

$\mathcal{H}$ is the observation operator, which links the control variables to the observed concentrations; it includes the CTM, space and time sampling, and other operations (e.g. averaging) required to compute the simulated equivalent of the assimilated data. The uncertainties in the observations $y$ together with those in the observation operator $\mathcal{H}$ and the uncertainties in the prior estimate of the control vector $x^{\text{b}}$ are assumed to have a Gaussian distribution and are thus characterized by their covariance matrices $\mathbf{R}$ and $\mathbf{B}$ respectively. The assumptions and practical way to define these matrices have been detailed by Fortems-Cheiney et al. (2021). The ratios between the prior error standard deviations in $\mathbf{B}$ and the prior estimates are set to 100 % for the CO emissions. This value of 100 % has already been chosen in the literature (Pétron et al., 2002; Kopacz et al., 2010; Yumimoto and Uno, 2006; Fortems-Cheiney et al., 2011, 2012, 2021). Even though annual CO emissions in western Europe may be well known, with uncertainties of 6 % according to Super et al. (2020), larger uncertainties could affect eastern Europe. Moreover, large uncertainties still affect bottom-up emission inventories at the $0.5°$ resolution: spatial disaggregation of the national-scale estimates to provide gridded estimates causes a significant increase in the uncertainty for CO (Super et al., 2020).

In contrast with Fortems-Cheiney et al. (2021), where they are set to 15 %, the ratios between the prior error standard deviations in $\mathbf{B}$ and the prior estimates are set to 50 % for the CO lateral conditions. Spatial correlations are built with exponentially decaying functions with an $e$-folding length of 50 km on land and on sea. Here, the covariance matrix $\mathbf{R}$ only takes into account the estimates of measurement errors reported in the MOPITT data sets. Indeed, the errors associated with the observation operators (in particular those associated with the chemistry-transport modelling with the CHIMERE configuration for Europe) are ignored since they are assumed to be much smaller than those associated with the MOPITT data. The minimum of the cost function $J$ is searched for with the iterative limited-memory quasi-Newton minimization algorithm M1QN3 (Gilbert and Lemaréchal, 1989). At each iteration, the computation of the gradient of $J$ relies on the adjoint of the observation operator, and in particular on the adjoint of CHIMERE. In the results presented in Sect. 3, the norm of the gradient of the cost function $J$ is reduced by more than 90 %, which indicates robust mathematical behaviour of the system.

The calculation of the uncertainty in the estimate of emissions from the inversion, known as "posterior uncertainty", is challenging when using a variational inverse system (Rayner et al., 2019): it is not done here.

## 3 Results

### 3.1 Comparison between simulated and assimilated CO concentrations

The MOPITT data and their prior simulated equivalents present similar spatial patterns for CO concentrations, with the lowest values over Spain (i.e. about 125 ppbv) and values higher than 200 ppbv over central Europe (over the Benelux, the Po Valley in Italy, north-western Germany, and southern Poland, Fig. 3). However, the prior simulation overestimates CO concentrations compared to the MOPITT super-observations, in particular over urban and industrial areas in central Europe, where the anthropogenic emissions are large (Fig. 2a).

It is interesting to note that global models have struggled, with a low bias in CO in the Northern Hemisphere, particularly in winter, compared to the MOPITT observations (Fortems-Cheiney et al., 2011; Stein et al., 2014). However, compared to these previous studies, we have used more recent MOPITT observations, and validation results for the Version 8 MOPITT CO products indicate reduced long-term bias drift, weaker-bias geographical variability, and smaller biases overall compared to Version 7 (Deeter et al., 2019). We have also used a more recent prior estimation of the CO emissions from the TNO-GHGco-v3 inventory, as it is based on recent EMEP/CEIP official country reporting for air pollutants. As model errors in long-range transport, diffusion, and chemistry linked to the hydroxyl radical OH and to NMVOCs (Strode et al., 2015; Gaubert et al., 2020) and coarse resolution (Valin et al., 2011) can all impact the inverse modelling of CO (Arellano et al., 2006; Fortems-Cheiney et al., 2011; Jiang et al., 2017; Zheng et al., 2019), we also used a chemical scheme describing the CO chemistry (including its secondary production through the oxidation and photolysis of hydrocarbons and its sink with OH, Sect. 2.1), and we have increased the spatial resolution of the transport model with a regional CTM. These different aspects can explain how our regional inversion does not highlight a low bias in the inventories, unlike past global inversion studies.

By design, the inversions bring the simulated CO concentrations closer to the MOPITT "surface" super-observations (Fig. 3d). The mean bias over the entire domain between the simulation and the MOPITT super-observations is reduced by about 2 %. Nevertheless, the corrections made to

the prior TNO-GHGco-v3 inventory are particularly large in areas where both CO emissions and the sensitivity of CO concentrations to the emissions are high (Figs. 2a, 4). For example, the posterior emissions reduce the mean bias between the simulated concentrations and MOPITT data by about 26 % over the Po Valley in Italy and over Benelux in January 2015 (Fig. 3d).

Nevertheless, it is worth stressing that the posterior simulation still presents positive biases compared to the observations (Fig. 3d). This can be explained by (i) large errors in the MOPITT super-observations that could reach 40 % (Fig. 5b) and (ii) the relatively weak sensitivity of the simulated concentrations to the local or regional emissions, as illustrated in Fig. 4.

## 3.2   Posterior CO emissions

This section focuses on the emissions from the 11-year CO inversion for the period 2011 to 2021. As the prior simulation overestimates CO concentrations compared to the MOPITT super-observations, the inversion applies negative increments to the prior emission estimates (Fig. 2b). These negative increments mainly occur in autumn and winter, even though there is a lower number of observations during these seasons compared to spring and summer. The highest increments are found over large cities and over industrial areas (Fig. 2b), where CO emissions are high (Fig. 2a). This shows the potential of MOPITT data to provide some information over areas with strong anthropogenic CO emissions.

The differences between the prior and posterior CO annual budgets for 30 European countries are shown in Table 2 for the year 2015. Annual budgets of the national emissions have decreased by about 1 %–11 % (Table 2). Similarly, the national and annual increments from 2011 to 2021 range between −1 % and 11 %. Overall, the posterior emission estimates are about 6.3 % lower than the prior emissions for the EU-27+UK area in 2015. This indicates that the European CO emissions could be slightly overestimated in the TNO-GHGco-v3 inventory.

The 2011–2021 inversion makes it possible to evaluate the trends and compare them to the ones indicated by the inventories over the decade. As our prior estimates of the emissions for 2020 and 2021 are set at the values for 2019 (see Sect. 2), trends of CO emissions are only computed from 2011 to 2019. This restriction avoids including the Covid-19 pandemic years.

The TNO-GHGco-v3 CO emissions show a decreasing trend over the EU-27+UK area from 2011 to 2019 (Fig. 1) of about $-2.5\ \%\ \mathrm{yr}^{-1}$ ($p = 9.5 \times 10^{-4}$). These decreasing trends are mainly driven by the transport sector (Zheng et al., 2019) with progressive pollution control on vehicles that has cut down European CO emissions (Crippa et al., 2016). Interestingly, the trends from 2011 to 2019 in the TNO inventory, based on the EMEP official reporting, exhibit some dispar-

**Table 2.** Difference between the CO annual emissions from the TNO-GHGco-v3 inventory used as a prior in this study and from the inversions, by country (%), in 2015.

| Country code | Difference between CO anthropogenic emission estimates from the inversions and TNO-GHGco-v3 (%) |
|---|---|
| ALB | −5.9 |
| AUT | −8.0 |
| BEL | −6.2 |
| BLR | −0.6 |
| CHE | −8.4 |
| DEU | −7.6 |
| DNK | −1.2 |
| ESP | −4.1 |
| FIN | −0.5 |
| FRA | −5.3 |
| GBR | −3.3 |
| IRL | −0.8 |
| ITA | −11.4 |
| LUX | −6.3 |
| NLD | −6.8 |
| NOR | −1.0 |
| PRT | −3.0 |
| SWE | −0.4 |
| BGR | −4.0 |
| CZE | −10.6 |
| EST | −0.3 |
| HRV | 8.94 |
| HUN | −7.1 |
| LTU | −0.7 |
| LVA | −0.4 |
| POL | −6.7 |
| ROU | −5.6 |
| SVN | −9.1 |
| SVK | −7.8 |
| UKR | −3.4 |
| EU-27+UK | −6.3 |

ities depending on the country, with for example a stronger decreasing trend over France than over Germany.

The posterior CO emissions display a very similar decreasing trend to the prior emissions over the EU-27+UK area (Fig. 1) of about $-2.2\ \%\ \mathrm{yr}^{-1}$ ($p = 2.2 \times 10^{-3}$). The main differences between the prior and posterior trends are found for the autumn and winter months, with a posterior trend of about $-1.9\ \%\ \mathrm{yr}^{-1}$ compared to the prior trend of about $-2.4\ \%\ \mathrm{yr}^{-1}$. Spatially, the differences are larger in Italy, in the Czech Republic, and in the Balkans than in the rest of Europe (Fig. 6). While the TNO-GHGco-v3 inventory shows significant decreasing trends in these regions, the posterior emissions appear to be stagnating, even with a non-significant increasing trend over parts of Italy. These areas benefit from the best MOPITT coverage, with the highest number of MOPITT super-observations (Fig. 5a). Con-

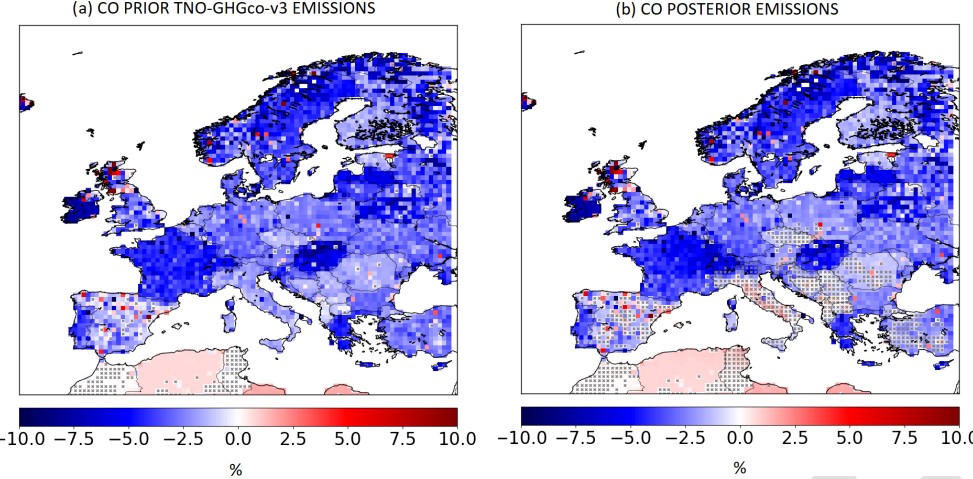

**Figure 6.** Trends of CO emissions from 2011 to 2019 **(a)** in the TNO-GHGco-v3 inventory and **(b)** in the posterior emissions ($\% \, \text{yr}^{-1}$). Crosses show pixels with an insignificant trend ($p$ value higher than 0.05).

sequently, the assimilation of MOPITT observations into the inversions attenuates the strong decreasing trend of the CO emissions in the TNO-GHGco-v3 inventory, particularly during autumn and winter.

Finally, there is no significant inter-annual variability from 2011 to 2019, neither in the prior CO emissions nor in the posterior CO emissions. Particular attention has consequently been paid to the possible detection of an inter-annual anomaly linked to the policies implemented in response to the Covid-19 pandemic in 2020.

### 3.3   Impact of Covid-19

Following the usual diagnostic in the literature to assess the change in air pollutant concentrations due to the Covid-19 policies, we characterize the impact of the Covid-19 policies in terms of a change in emissions budgets from April 2019 to April 2020. Most of the European countries implemented lockdown policies in April after a progressive implementation of the national lockdowns from 9 March 2020 (Italy) to 23 March 2020 (United Kingdom, UK). The change from April 2019 to April 2020 potentially includes variations associated with drivers of the usual emission processes (e.g. changes in temperature from 2019 to 2020), but, as indicated above, the typical inter-annual variations in both the prior and posterior estimates are relatively small. Since the prior estimates for 2020 and 2019 are identical (see Sect. 2), we actually analyse the impact of the Covid-19 policies in terms of differences of increments provided by the inversions to these prior estimates between April 2019 and April 2020. Overall, a much smaller lockdown-driven impact is expected for CO than for $NO_2$, particularly because of smaller contributions from lockdown-affected sources (Clark et al., 2021).

At the European scale, the CO posterior emission estimates derived from the MOPITT data decrease by about

**Table 3.** Difference between the CO posterior emissions in April 2020 and in April 2019, by country (%).

| Country code | Difference between CO posterior emissions estimates in April 2020 and in April 2019 (%) |
|---|---|
| BEL | −5.6 |
| CHE | −3.5 |
| DEU | −7.3 |
| FRA | −1.2 |
| GBR | −0.4 |
| ITA | −3.5 |
| LUX | −5.4 |
| NLD | −7.7 |
| EU-27+UK | −1.3 |

−1.3 % in April 2020 compared to April 2019 (Table 3). This decrease is lower than the estimates of about −4.7 %, −6.4 %, −7.6 %, and −8.2 % of respectively Guevara et al. (2023), Doumbia et al. (2021), Forster et al. (2021), and the officially reported emissions from EMEP/CEIP (CEIP, 2022).

Nevertheless, as shown in Fig. 7, the inversions lead to a higher decrease in CO emissions over the areas where the anthropogenic emissions are usually large, and particularly over industrial basins such as over the Benelux and Rhine–Ruhr Valley, where the decrease reaches −8 %, and over the Po Valley, where it reaches −10 % in April 2020 compared to April 2019 (Fig. 7c).

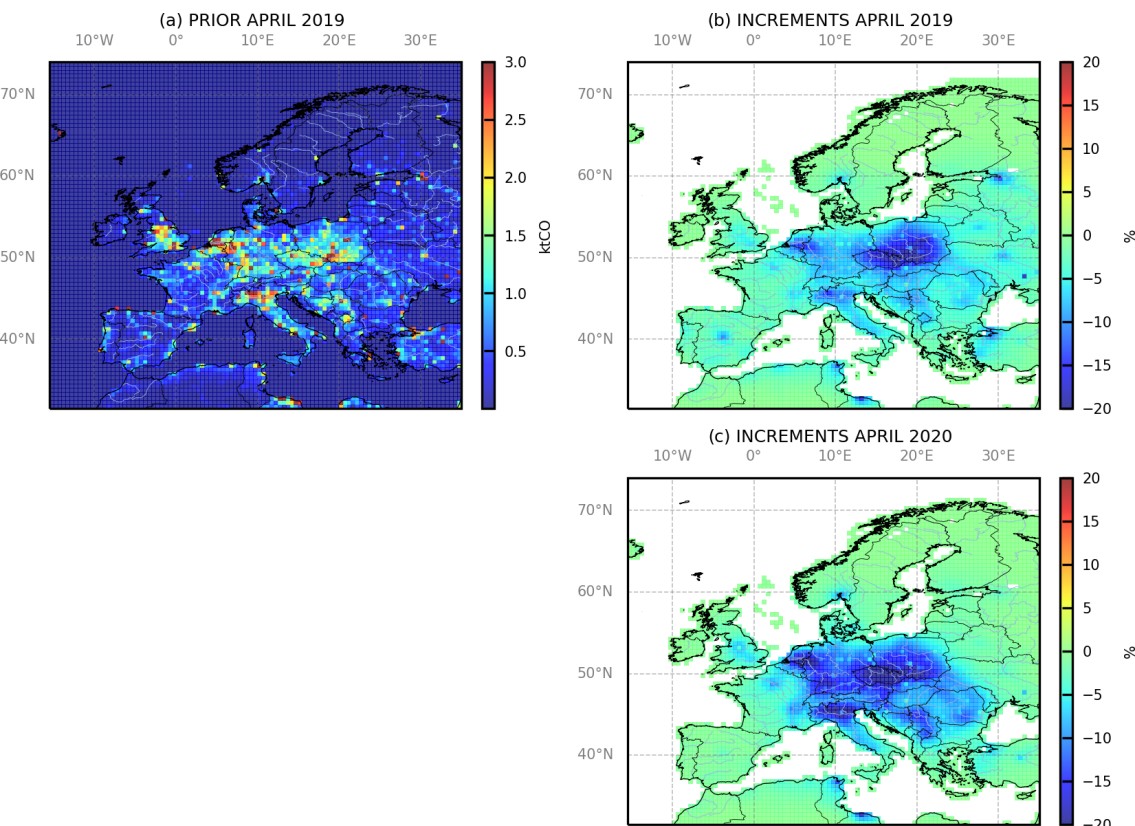

**Figure 7. (a)** Emissions estimated by the TNO inventory in April 2019 (kt CO per month). Increment provided by the inversions in **(b)** April 2019 and **(c)** April 2020 (%).

## 4   Discussion and conclusion

The CIF, coupled to the regional chemistry-transport model CHIMERE and its adjoint, together with the satellite CO MOPITT data have been used to estimate 11 years from 2011 to 2021 of European CO emissions. The analysis of the inversion results reveals the challenges associated with the inversion of CO emissions at the regional scale over Europe. Annual budgets of the national emissions have decreased by about 1 %–11 % over the decade and over Europe. These decreases are mainly due to negative corrections during autumn and winter.

The posterior CO emissions display a very similar decreasing trend to the prior emissions over the EU-27+UK area, with a trend of about $-2.2\,\%\,\mathrm{yr}^{-1}$ showing a general consistency with reported anthropogenic emissions. This trend is slightly lower than in the prior emissions. The assimilation of the MOPITT observation in the inversions indeed attenuates the decreasing trend of the CO emissions in the TNO inventory over areas benefiting from the highest number of MOPITT super-observations (particularly over Italy and over the Balkans), and particularly in autumn and winter.

The posterior simulation still presents positive biases compared to the observations. The minimization algorithm of the inversion appears to converge correctly with the constraints used in practice. Therefore, these residual positive biases can be mainly explained for a large part by the large errors associated with the observations in our inversion framework. As discussed in Sect. 2.2, our derivation of the error associated with each super-observation is conservative. Other indices support this assumption. In particular, the $\chi^2$ diagnostic (Ménard and Chang, 2000) is significantly lower than 1. This indicates that the **B** and **R** matrices used here to characterize the prior and observation errors likely overestimate the amplitudes of these errors (Ménard and Chang, 2000). However, even when assuming that the observation errors would only consist of random noise uncorrelated in space and setting the error in the super-observation to that of the average of the number of observations (nbobs) in the model grid cells, i.e. of the order of $\frac{1}{\sqrt{(\mathrm{nbobs})}}$ times the observation error for individual observations, the impact would be moderate since nbobs is generally equal to 2 to 3. Actually, the set-up of the **B** matrix is also rather conservative, and the balance between the two errors in the set-up of the **B** and **R** matrices may be relatively good. Therefore, the lack of fit to the observations in these inversions could be associated with the large retrieval error corresponding to the MOPITT product. The robustness of the inversions would still benefit from a refinement of our

configuration of the **R** matrix, which would lead to a better fit to the observations. First, we should probably investigate the components of the retrieval errors which are distributed along with the MOPITT product. Gaubert et al. (2023) indicate that, when applying the averaging kernel, the smoothing error can be ignored and the weight of this component is significant. The revision of our conservative assignment of the observation errors to the super-observation would be more challenging. It would require good knowledge of the respective weight of the random noise (without spatial correlation) and the systematic errors (with spatial correlations) in the total retrieval errors as well as good knowledge of the typical correlation length scales of the systematic errors, but we lack insights regarding this. The use of notional assumptions (as for the characterization of the model error) may still represent a sensible trade-off and allow for an improved assimilation of the observations. Finally, a refinement of the inversion strategy may also support a better fit to the observations. In particular, under the assumptions that uncertainties in the control variables have a Gaussian distribution, the control of the logarithm of the emissions rather than the scaling factor for these emissions may better correspond to our CO inversion problem, in which CO emissions are necessarily positive but in which these emissions would have to be strongly decreased. In contrast to the Gaussian characterization and the spatial correlation of the uncertainties in the emissions, the Gaussian characterization and the spatial correlation of the uncertainties in the logarithm of the emissions could increase the flexibility for large local corrections of the emissions. The current characterization of the uncertainties in the CO emissions using a Gaussian distribution may actually contribute to the limitation of the fit to the observations.

The small corrections of the CO emissions at national scales by the inversion can be attributed, first, to the general consistency between the TNO-GHGco-v3 inventory and the satellite data. However, analysis of specific patterns such as the impact of the Covid-19 crisis reveals that it can also be seen as a lack of observation constraints to adjust the prior estimate of the emissions. The large errors associated with the observations in our inversion framework and the lack of data over large parts of Europe are definitely some sources of limitation on the observational constraint.

However, in a more general way, this questions the ability to exploit large-scale variations in the CO satellite data to constrain regional-, national-, and continental-scale budgets of the emissions. Emission hotspots generate a relatively strong local signal, which is much better caught and exploited by the inversions than the larger-scale signals despite the moderate spatial resolution of the MOPITT data. This is why the corrections of these hotspot emissions are stronger and more convincing than the corrections of the national- and continental-scale emissions, as shown by the analysis of the impact of Covid-19 policies. Accurate monitoring of the national anthropogenic CO emissions will likely rely more on the aggregation of local emission monitoring data than on the processing of large-scale variations in the CO fields. The former requires modelling and inversion systems at spatial resolutions finer than those used here as well as satellite images at high spatial resolutions. The CO images of the TROPOMI instrument on board the Sentinel-5P mission with a $5.5\,\text{km} \times 7\,\text{km}$ resolution since August 2019 should be well suited for such a perspective. The large increase in the number of observations with this mission is expected to increase the capabilities to monitor CO emissions and to address air-quality-related emissions at the national to sub-national scales.

**Data availability.** MOPITT Version 8 products are freely available through NASA's EarthData portal at https://earthdata.nasa.gov/ TS3 (Deeter et al., 2019). The TNO-GHGco-v3 inventory (Super et al., 2020) is available upon request from TNO (contact: Hugo Denier van der Gon, hugo.deniervandergon@tno.nl). The CHIMERE code is available here: http://www.lmd.polytechnique.fr/chimere/ TS4 (Menut et al., 2013; Mailler et al., 2017). The CIF inversion system (Berchet et al., 2021) is available at http://community-inversion.eu/ TS5.

**Author contributions.** AFC and GB conceptualized the study and carried out the results analysis. AFC carried out the inversions. EP, RP, AB and IP developed the CIF inversion system, including preprocessing for fluxes and satellite observations. HDvdG and SD provided the TNO-GHGco-v3 inventory used as prior emissions in this study. All the co-authors contributed to writing the manuscript.

**Competing interests.** The contact author has declared that none of the authors has any competing interests.

**Disclaimer.** Publisher's note: Copernicus Publications remains neutral with regard to jurisdictional claims made in the text, published maps, institutional affiliations, or any other geographical representation in this paper. While Copernicus Publications makes every effort to include appropriate place names, the final responsibility lies with the authors.

**Acknowledgements.** We acknowledge the NCAR MOPITT group for the production of the CO retrievals. A large part of the development and analysis was conducted in the framework of H2020 VERIFY funded by the European Commission Horizon 2020 research and innovation program under agreement no. 776810. We wish to thank all the persons involved in the preparation, coordination and management of this project. This study has also received support from the H2020 COCO2 project under agreement no. 958927, from the French ANR project ARGONAUT under grant agreement no. ANR-19-CE01-0007 and from the French AQA-CIA project LOCKAIR under grant agreement no. 2162D0010. This work was supported by the CNES (Centre National d'Etudes Spatiales) in the framework of the TOSCA ARGOS project. This work was granted access to the high-performance computing

(HPC) resources of TGCC under the allocations A0100102201 to A0140102201 made by GENCI. Finally, we wish to thank Julien Bruna (LSCE) and his team for computer support.

**Financial support.** This research has been supported by the Horizon 2020 Framework Programme (grant nos. 776810 and 958927), the French ANR project ARGONAUT (grant agreement no. ANR-19-CE01-0007), the French AQACIA project LOCKAIR (grant agreement no. 2162D0010), and the CNES (Centre National d'Etudes Spatiales, TOSCA ARGOS project). TS6

**Review statement.** This paper was edited by Qiang Zhang and reviewed by two anonymous referees.

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

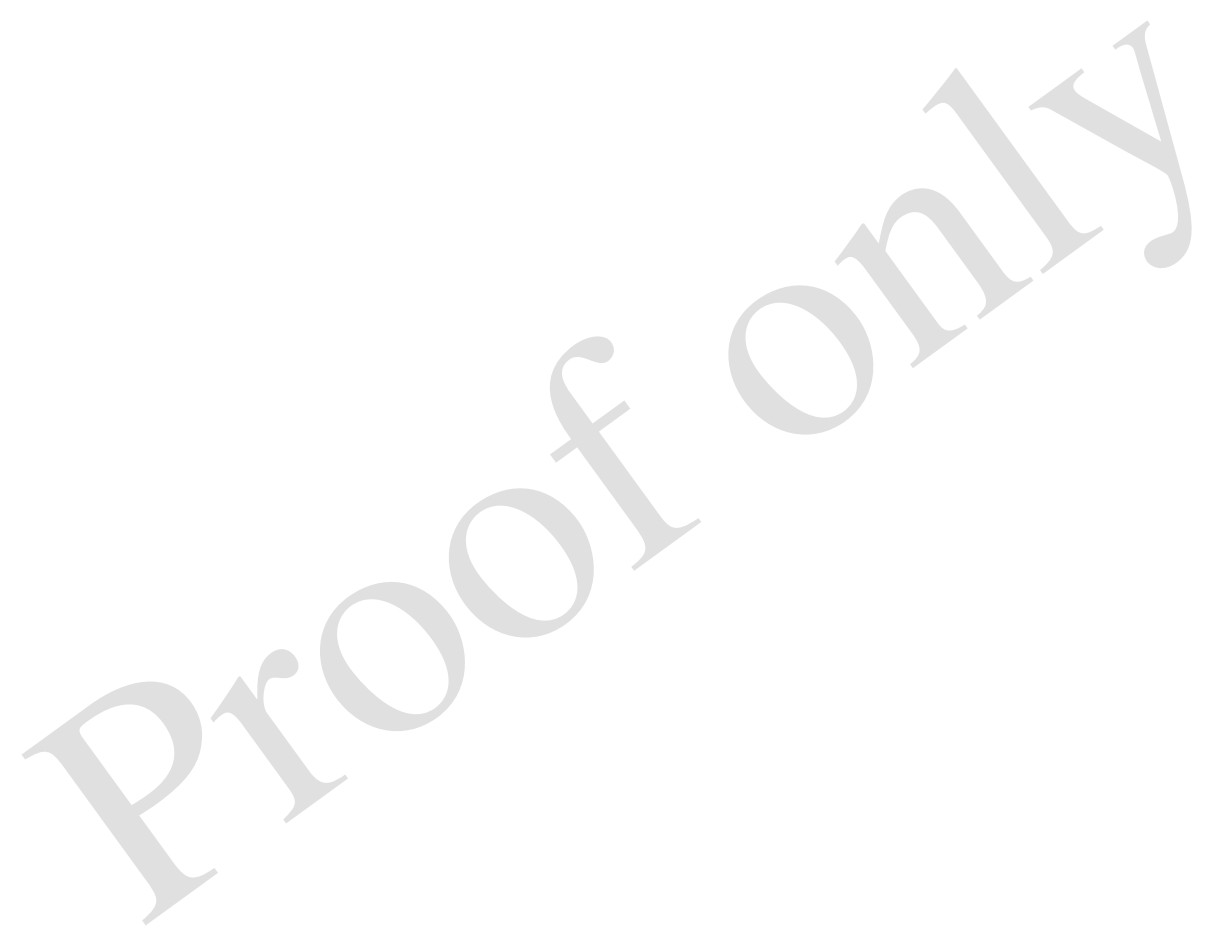

## Remarks from the typesetter