# Peer review of "The CO anthropogenic emissions in Europe from 2011 to 2021: insights from the MOPITT satellite data."

_EGUsphere, 2023_

## Author Comment (AC1)

**Reviewer #1**

Review of The CO anthropogenic emissions in Europe from 2011 to 2021: insights from the MOPITT satellite data by Audrey Fortems-Cheiney et al.

This study could provide interesting insights on an emission inversion framework for CO at the European scale. Posterior emissions are used to verify trends.

**We wish to thank the referee for his/her helpful comments. His/her full comments are copied hereafter in normal black font, and our responses are inserted in between in bold font.**

There is a lack of bibliographic references, which leads to misleading statements in the introduction and probably for some of the design of the study and the choice of parameters used in the inversions. There has been CO emission inversions at the regional scales, for instance Jiang et al. (2015) and Qu et al., (2022) performed regional MOPITT inversions at the grid cell level at 0.5° × 0.667°.

**We added references for the study of CO at the global scale in the introduction: « Over the last two decades, the space-borne Measurement of Pollution in the Troposphere ... have revolutionized our ability to map CO concentrations and to understand the trends and the spatio-temporal variability of its concentrations and emissions(Arellano et al., 2006;Chevallier et al., 2009; Jones et al., 2009; Kopacz et al., 2010; Jiang et al., 2011; Fortems-Cheiney et al., 2011; Hooghiemstra et al., 2012; Miyazaki et al., 2015; George et al., 2015; Yin et al., 2015; Jiang et al., 2017; Zheng et al., 2018; Buchholz et al., 2021; Gaubert et al., 2023).”**

**We have also added the references for the study of CO at the regional scale: « In the past decade, CO regional scale inversions based on the MOPITT data covered CO emissions in North America (Jiang et al., 2015) and East Asia (Qu et al., 2022). To our knowledge, there has been only a few studies covering the European CO emissions based on satellite observations (Konovalov et al., 2016; Fortems-Cheiney et al., 2021), this continent being more challenging for regional scale inversions of the CO anthropogenic emissions owing to weaker CO signal (Konovalov et al., 2016).»**

**We have also changed the sentence: « The ability of regional inverse systems to quantify CO anthropogenic emission budgets at the national and monthly to annual scales in Europe from the MOPITT TIR-NIR satellite observations has not been assessed yet.»**

There are two main concerns:

1.      MOPITT errors reported on Fig. 3c (30 to 40 %) seems to be larger than usual.

**Indeed, the MOPITT errors appear to be larger over Europe than over other parts of the globe, such as over Eastern China (see the figure below).**

[Figure]

*Averages of the errors associated with theCO MOPITT super observations a) over Europe in January 2015 as in Figure 5b and b) over Eastern China in January 2019, in %.*

Regarding this topic, it is important to note that studies in the literature often use MOPITT "super-observations" within relatively large model grid, i.e. aggregated observations based on the average of all the observation within the model grid cells. The corresponding observation error, in principle, should be smaller than that of the individual observations (typically by a factor of 1/sqrt(nobs)when deriving super-observations as the mean of the observations, and if assuming that the errors in the different retrieval within a grid cell are totally independent from each other). However, here, when defining such super-observations as the observation whose value is the median of the ensemble of observation within a 0.5°x0.5° grid cell of the CTM, and within the CTM physical time steps, we assign the observation error associated to this observation to the super-observation, which is implicitly a conservative observation error derivation. We assume that a large part of the errors on the individual retrievals bear spatial correlations between the different retrievals at the scale of the CHIMERE grid cells (Deeter et al., 2019).

In any case, the typical number of MOPITT observations per CTM grid cell is often about 1 or 2or 3 observations. Cases when it is 4 to 6 are relatively rare, so that the choice to decrease the error on the super-observation as a function of the number of observation or not do not imply large changes in most of the cases.

We have changed the sentences: « In order to associate the super-observations to a real AK, the super-observations have been taken as the individual observation corresponding to the value of the median of the MOPITT concentrations within the 0.5°x0.5°grid-cell of the CTM and within the CTM physical time steps (about 5-10 min). The AK and the uncertainty associated to this individual value are then used to define the AK and uncertainty for the « super-observation ». In principle, the observation error associated to such a median value should be smaller than the error associated to individual observation, but, here, we keep the error for the individual observation used to define the super-observation as a conservative estimate of the super-observation error. The super-observations therefore do not have a smaller error than the individual observations. »

This is concerning because the main result of the paper indicates a lack of convergence of the emission optimization "The posterior simulation still presents positive biases compared to the observations, which can be partly explained by large errors in the MOPITT observations".

**Yes, we agree that the lack of convergence can be partly explained by the large errors in the MOPITT observations. This is now better detailed in the conclusion. However, we cannot push the system to overfit these data and we must follow the best knowledge of the level of errors in the Bayesian inversion framework.**

There are techniques to adjust the errors in order to reach convergence and at least an evaluation of the model data mismatch could be presented.

**We already indicate numbers demonstrating that our system reaches convergence (reduction of the norm of the gradient of the cost function by more than 90%)and we have checked that the reduction of the cost function (and its observational component) hardly evolve  in the last iterations of the inversions (i.e. that the algorithm has converged well). The model data mismatch is also already presented in Figure 3.**

The MOPITT data reports both the measurement error covariance and the smoothing error covariance in addition to the often use total retrieval error covariance. In this study, the averaging kernels (smoothing) are applied in the observation operator (as it should be) to estimate the columns, only the measurement error covariance should then be included, as done in Gaubert et al., (2023).

**We acknowledge that we are not familiar with this type of treatment for the retrieval error. We are accustomed to following the recommendations of the MOPITT user's guides and to our knowledge, there is no mention of using only measurement error in the recommendations of these user's guides so far. Here, we have used the estimated errors (i.e., uncertainties) available in the error field (second element) of the "Retrieved CO Surface Mixing Ratio" variables of the MOP02 files, including a smoothing error and the instrumental noise, as indicated in a MOPITT user's guide (MOPITT V7 Level 2 Data Quality Summary, 2016).**
**Nevertheless, we take this information into account. We have added this information in the conclusion: "The robustness of the inversions would still benefit from a refinement of our configuration of the R matrix which would lead to a better fit to the observations. First, we should probably investigate the components of the retrieval errors which are distributed along with the MOPITT product. Gaubert et al., (2023) indicate that when applying the averaging kernel, the smoothing error could be ignored and that the weight of this component is significant.»**

The authors should at least consider showing some statistics on the convergence  (e.g., chi-square) of the assimilation for the entire run and adjust the errors accordingly.

**As mentioned above, we already show some statistics on the convergence with the reduction of the norm of the gradient of the cost function by more than 90%, which indicates a robust mathematic behavior of the system.**

**The uncertainties in the observations and the uncertainties in the prior estimate of the control vector are characterized by their covariance matrices R and B. The matrices B and**

R are generally derived from expert knowledge based on studies on the performances of the atmospheric and process models (Fortems-Cheiney et al., 2021). The derivation of the B covariance matrix is a complicated task and there is still a critical lack of knowledge on the amplitude and spatio-temporal patterns of uncertainties in anthropogenic emissions of pollutants and GHG (Super et al., 2020).

We have already detailed above our rationale and computations for the derivation of the R matrix. The error standard deviations assigned to the prior CO emissions in B at 1-day and 0.5$^\square$ resolution are 100 %. We have added sentences in the text explaining this choice: « This value of 100 % has already been chosen in the literature (Pétron et al. 2002 ; Yumimotoa and Uno 2006, Kopacz et al. 2010 ; Fortems-Cheiney et al. 2011 ; Fortems-Cheiney et al. 2012 ; Fortems-Cheiney et al., 2021).Even though annual CO emissions in western Europe may be well known, with uncertainties of 6 % according to Super et al. (2020), larger uncertainties could affect eastern Europe. Moreover, large uncertainties still affect bottom-up emission inventories at the 0.5$^\square$ resolution: spatial disaggregation of the national-scale estimates to provide gridded estimates causes a significant increase in the uncertainty for CO (Super et al., 2020) ».

The $\chi 2$ value is used to diagnose balance between actual errors and estimated errors. Our $\chi 2$ is of about 0.2 for January 2015.When the $\chi 2$ value is smaller than the ideal value of 1, it suggests overestimated background error covariance or observation errors. This $\chi 2$ may therefore confirm that our set-up of the uncertainties in the covariance matrices R are too large (see the discussion above regarding the assignment of the observation error for super-observations). However, the typical factor of decrease of the observation error for super-observation when accounting for the noise component of the individual retrieval errors may hardly explain that the $\chi 2$ is about 0.2. Actually, we also have an important level of prior uncertainty at the model grid cell and 1-day scale (100%), i.e. the balance between the prior and observation uncertainties is not necessarily misrepresented. The robustness of the interpretation of the $\chi 2$ diagnostic can thus hardly be used to correct the observation errors, for which we have tried to assign a sensible level based on the reported retrieval errors.

As mentioned above, we now indicate in the conclusion that our derivation of the error associated with each super-observation could be conservative, and we discuss the potential and challenges associated to the improvement of the assignment of observation errors for individual observations (Gaubert et al., 2023) and for super-observations:« The posterior simulation still presents positive biases compared to the observations. The minimization algorithm of the inversion appears to converge correctly with the constraints used in practice. Therefore, these residual positive biases can mainly explained by the large errors associated to the observations in our inversion framework. As discussed in Section 2.2, our derivation of the error associated with each super-observation is conservative. Other indices support this assumption. In particular, the $\chi 2$ diagnostic (Ménard and Chang, 2000) is significantly lower than 1. This indicates that the B and R matrices used here to characterize the prior and observation errors likely overestimates the amplitude of these errors (Ménard and Chang, 2000).  However, even if assuming that the observation errors would only consist in random noise uncorrelated in space and setting the error on the super-observation as that of the average of the number of observations nbobs in the model

grid cells, i.e. of the order of 1/sqrt(nbobs) times the observation error for individual observations, the impact would be moderate since nbobs is generally equal to 1 to 3. Actually, the set-up of the B matrix is also rather conservative, and the balance between the two errors in the set-up of the B and R matrices may be relatively good. Therefore, the lack of fit to the observations in these inversions could be associated to the large retrieval error corresponding to the MOPITT product. The robustness of the inversions would still benefit from a refinement of our configuration of the R matrix which would lead to a better fit to the observations. First, we should probably investigate the components of the retrieval errors which are distributed along with the MOPITT product. Gaubert et al., (2023) indicate that when applying the averaging kernel, the smoothing error could be ignored and that the weight of this component is significant. The revision of our conservative assignment of the observation errors to the super-observation would be more challenging. It would require a good knowledge of the respective weight of the random noise (without spatial correlation) and of the systematic errors (with spatial correlations) in the total retrieval errors, as well as a good knowledge of the typical correlation length scales of the systematic errors, while we could lack of insights regarding this. The use of notional assumptions (as for the characterization of the model error) may still represent a sensible trade-off and allow for an improved assimilation of the observations. »

Fortems-Cheiney, A., F. Chevallier, I. Pison, P. Bousquet, S. Szopa, M. N. Deeter, and C. Clerbaux (2011), Ten years of CO emissions as seen from Measurements of Pollution in the Troposphere (MOPITT), *J. Geophys. Res.*, 116, D05304, doi:10.1029/2010JD014416.

Fortems-Cheiney, A., Chevallier, F., Pison, I., Bousquet, P., Saunois, M., Szopa, S., Cressot, C., Kurosu, T. P., Chance, K., and Fried, A.: The formaldehyde budget as seen by a global-scale multi-constraint and multi-species inversion system, Atmos. Chem. Phys., 12, 6699–6721, https://doi.org/10.5194/acp-12-6699-2012, 2012.

Kopacz, M., Jacob, D. J., Fisher, J. A., Logan, J. A., Zhang, L., Megretskaia, I. A., Yantosca, R. M., Singh, K., Henze, D. K., Burrows, J. P., Buchwitz, M., Khlystova, I., McMillan, W. W., Gille, J. C., Edwards, D. P., Eldering, A., Thouret, V., and Nedelec, P.: Global estimates of CO sources with high resolution by adjoint inversion of multiple satellite datasets (MOPITT, AIRS, SCIAMACHY, TES), Atmos. Chem. Phys., 10, 855–876, https://doi.org/10.5194/acp-10-855-2010, 2010.

Ménard, R., and L. Chang, 2000: Assimilation of Stratospheric Chemical Tracer Observations Using a Kalman Filter. Part II: χ2-Validated Results and Analysis of Variance and Correlation Dynamics. *Mon. Wea. Rev.*, **128**, 2672–2686, https://doi.org/10.1175/1520-0493(2000)128<2672:AOSCTO>2.0.CO;2.

Pétron, G., C. Granier, B. Khattatov, J.-F. Lamarque, V. Yudin, J.-F. Müller, and J. Gille, Inverse modeling of carbon monoxide surface emissions using Climate Monitoring and Diagnostics Laboratory network observations, *J. Geophys. Res.*, 107(D24), 4761, doi:10.1029/2001JD001305, 2002.

Yumimotoa K, Uno I. Adjoint inverse modeling of CO emissions over Eastern Asia using four-dimensional variational data assimilation. Atmos. Chem. Phys. 2006; 40: 6836–6845. 10.3402/tellusb.v64i0.19047.

2.    Using a spatial correlations e-folding length of 50 km is effectively forcing the system to constrain hotspots only. Qu et al., (2022) used correlation lengths that varied by sectors with a 100 km to 200 km range. Only point sources from the energy sectors were considered to be at scales smaller than 100 km.

**With our set-up, the inversions system is free, in principle, to homogeneously correct large regions to match large scale gradients in the satellite data, which cover the European domain relatively well; putting larger spatial correlations in the prior error covariance matrix would smooth the corrections of the inversion but it has to be consistent with the actual structures of error correlations in the gridded inventories and we are not aware of analysis demonstrating that these structures have long typical lengths.**

**On the opposite, we think that isotropic correlations decreasing with distance are not good representations of the error on anthropogenic emissions, even for large sectors of activity; considering the heterogeneity of anthropogenic emissions favors a more conservative view by limiting extrapolation via the matrix B (see Super et al., 2023 for an analysis of the uncertainties in the gridded emissions of $CO_2$).**

**Our results show that the inversions do not really filter large scale variations of CO in the data associated to the emissions, the MOPITT data providing constraints over hotspots only. Such a result should not be blurred by artificially extrapolating the information on the emission hotspot to the rest of the countries via spatial correlations in the prior error covariances.**

Super, I., Choulga, M. and Hohenberger, T.: PED uncertainty 2018and uncertainties based on Monte Carlo simulation using the emission model from D2.5 v2, D2.7 COCO2 Deliverable, https://www.coco2-project.eu/sites/default/files/2023-11/CoCO2-D2-7-V1-0.pdf

Ma et al. (2019) and Gaubert et al. (2020) considers larger correlation lengths of 600 and 500 km on the basis that emission inventories are constructed at the province level (China) or at the scale of entire countries.

**In general, the emission inventories are indeed often constructed at the scale of country and the robustness of the budgets are considered maximum at the national scale. However, such national budgets are then disaggregated in space and this spatial disaggregation, in principle, creates uncertainty and negative correlations at sub-national scale. Furthermore, the disaggregation follows average activity maps which generally lack of granularity, generating complex and heterogeneous spatial structures of the uncertainties. Errors on average emission factors by detailed sectors of activity, create positive correlations across the countries and provinces, but the correlations within the countries associated to the actual local emission factors are complex, the errors in average emission factors generate varying correlations across countries due to highly varying situations between different pairs of countries, and the combination between these correlations and the error structure associated to the spatial disaggregation of the emissions results in complex error correlation features, especially when working with aggregated sectors. Again, this pushes for a conservative modeling of the spatial correlation to avoid abusive extrapolation of the**

information from the constraint of the satellite data on emission hotspots(e.g. from traffic in large urban areas to major road axis or to traffic in small towns).

While they are not inverting the CO emissions, Inness et al., 2023 show a global mean horizontal correlation length of 125 km at the surface level (Figure 1). This is important when the objective of the paper is to assessed "the ability of regional inverse systems to quantify CO budgets at the national scale from the MOPITT TIR-NIR satellite observations" and that the corrections of the hot spots are more "convincing".

**Inness et al. 2023 do not derive this correlation length from an analysis of the structures of errors in emission inventories. The use of isotropic error correlations decreasing with distance with long length-scales is a kind of tradition from atmospheric transport inversion cases with poor observation coverage and targeting relatively diffuse flux fields (such as natural greenhouse gases fluxes), but it may not be appropriate to tackle the anthropogenic emissions and to spatialized inventories, disaggregated from national budgets. This explains why "fossil fuel data assimilation" systems where the inversions controls parameters of emission models rather than emission maps become increasingly appealing(Kaminski et al., 2022).Furthermore, the good coverage from satellite observation should allow not relying on such simple correlation model to ensure that the top down constraint in inversions is significant. Working with short correlations emphasize the direct observational constraint in the inversion, and the spatial coverage of the MOPITT observations could have allowed, in principle, a direct control of the national scale budgets, even though the results demonstrate it is currently not the case.**

Kaminski, T *et al* : Assimilation of atmospheric $CO_2$ observations from space can support national $CO_2$ emission inventories, *Environ. Res. Lett.*17 014015, DOI 10.1088/1748-9326/ac3cea, 2022.

Minor comments:

Check that the figures appear in the same order in the text.

**This has been done.**

Figure 5's colorbar indicates "ppb" while the maps show the number of observations.

**This has been corrected.**

References:

Gaubert, B., Emmons, L. K., Raeder, K., Tilmes, S., Miyazaki, K., Arellano Jr., A. F., Elguindi, N., Granier, C., Tang, W., Barré, J., Worden, H. M., Buchholz, R. R., Edwards, D. P., Franke, P., Anderson, J. L., Saunois, M., Schroeder, J., Woo, J.-H., Simpson, I. J., Blake, D. R., Meinardi, S., Wennberg, P. O., Crounse, J., Teng, A., Kim, M., Dickerson, R. R., He, H., Ren, X., Pusede, S. E., and Diskin, G. S.: Correcting model biases of CO in East Asia: impact on oxidant distributions during KORUS-AQ, Atmos. Chem. Phys., 20, 14617–14647, https://doi.org/10.5194/acp-20-14617-2020, 2020.

Gaubert, B.; Edwards, D.P.; Anderson, J.L.; Arellano, A.F.; Barré, J.; Buchholz, R.R.; Darras, S.; Emmons, L.K.; Fillmore, D.; Granier, C.; et al. Global Scale Inversions from MOPITT CO and MODIS AOD. Remote Sens. 2023, 15, 4813. https://doi.org/10.3390/rs15194813

Ma, C., Wang, T., Mizzi, A. P., Anderson, J. L., Zhuang, B., Xie, M., & Wu, R. (2019). Multiconstituent data assimilation with WRF-Chem/DART: Potential for adjusting anthropogenic emissions and improving air quality forecasts over eastern China. Journal of Geophysical Research: Atmospheres, 124, 7393–7412. https://doi.org/10.1029/2019JD030421

Inness, A., Aben, I., Ades, M., Borsdorff, T., Flemming, J., Jones, L., Landgraf, J., Langerock, B., Nedelec, P., Parrington, M., and Ribas, R.: Assimilation of S5P/TROPOMI carbon monoxide data with the global CAMS near-real-time system, Atmos. Chem. Phys., 22, 14355–14376, https://doi.org/10.5194/acp-22-14355-2022, 2022.

Jiang, Z., Jones, D. B. A., Worden, J., Worden, H. M., Henze, D. K., and Wang, Y. X.: Regional data assimilation of multi-spectral MOPITT observations of CO over North America, Atmos. Chem. Phys., 15, 6801–6814, https://doi.org/10.5194/acp-15-6801-2015, 2015.

Qu, Z., Henze, D. K., Worden, H. M., Jiang, Z., Gaubert, B., Theys, N., & Wang, W. (2022). Sector-based top-down estimates of NOx, SO2, and CO emissions in East Asia. Geophysical Research Letters, 49, e2021GL096009. https://doi. org/10.1029/2021GL096009

---

## Author Comment (AC2)

**Reviewer #2**

In their paper Fortems-Cheiney et al. present the assimilation of MOPITT CO data over Europe and discuss emission trends for a 10-year period. The paper is well written and well structured. I am in favour of publishing these results, but also have several requests for clarifications and minor adjustments, as listed below.

**We wish to thank the referee for his/her helpful comments. His/her full comments are copied hereafter in normal black font, and our responses are inserted in between in bold font.**

l 24: ..here, as well as satellite ..

**This has been corrected.**

l 98: "CO emissions from fires, .. not taken into account" It is difficult for me to understand the distribution shown in Figure 4. In particular the high values in Eastern Europe. Is this linked to fires or something else? is this inflow of CO through the Eastern domain boundary?

**These high values in Eastern Europe are not due to fires or to inflow of CO through the Eastern domain boundary. This is mainly due to the application of the AK and of the MOPITT prior profiles to the CHIMERE simulations, as shown in the figure below.**

**This figure now replaces Figure 3 and we have added the following text to Section 2.2: «The resulting monthly means of the MOPITT super-observations and their simulated equivalents for CO average surface concentrations in January 2015 are respectively illustrated in Figure 3b and in Figure 3c. The spatial patterns of the CO concentrations are very different if using directly the CO columns up to 900hPa (Figure 3a) or if the MOPITT AK and prior profiles are applied (i.e., somehow, by projecting the model column into the MOPITT retrieval space; Figure 3b), particularly in Central, Eastern and Northern Europe. It bears evidence that the MOPITTAK and prior profiles have a strong impact on the CO concentrations over these regions. »**

[Figure]

*Figure 3. Averages of the CO concentrations between the surface and 900 hPa,a) simulated by CHIMERE using the prior TNO-GHGco-v3 anthropogenic emission estimate without applying the MOPITT AK and prior profiles, b) corresponding to the MOPITT "surface super observations"in the CHIMERE grid, c) simulated by CHIMERE using the prior TNO-GHGco-v3 anthropogenic emission estimate applying the MOPITT AK and prior profiles, in ppbv. d) Ratios of the posterior and prior biases between monthly mean surface concentrations from CHIMERE and the MOPITT super-observations, at the 0.5°x0.5° grid-cell resolution, in January2015. All ratios lower than 1, in blue, demonstrate that posterior emission estimates improve the simulation compared to the prior ones.*

l 100: "CO biogenic emissions are assumed to be negligible and are not taken into account." But Table 1 mentions that MEGAN is used, and this seems to contradict this statement. Please explain.

**As stated in the introduction, CO has a major role in atmospheric chemistry. CO concentration is influenced by reactions with other species such as hydroxyl radical (OH), non-methane volatile organic compounds (NMVOCs) or tropospheric ozone (O$_3$) whose concentration must be accurately represented.**

**For this objective, and as explained in lines 115-120, the chemical scheme MELCHIOR-2 used here for the simulation of CO concentrations needs emissions from other species, such as non-methane volatile organic compounds (NMVOCs) or nitrogen oxides (NO$_x$). Biogenic NO$_x$ and NMVOC emissions, in particular emissions of isoprene and some other hydrocarbons from vegetation, are obtained from the MEGAN model.**

Also, Fortems-Cheiney (2021) contains a figure 3 which shows the importance of biogenic + anthro emissions. I could not connect this to Fig. 4 which indicates just a very small impact of the emissions.

**Indeed, but the figure in Fortems-Cheiney et al. (2021) only presents results for one week and for another period.**

l 113: Fig 2a is mentioned, but figure 1 is only referenced in line 227. Please change the order of the figures.

**Figure 1 is now mentioned before Figure 2.**

Sec 2.1: Please explain how emissions are distributed on the vertical model layers for the different sectors.

**The TNO-GHGco inventory combines emissions from area sources, injected at the surface in the model, and from point sources. Emissions from point sources, mainly from the energy production and the industrial sector, are distributed on the vertical model layers depending on typical injection height provided in the TNO inventory, based on Bieser et al. (2011).All these information have been added in the text in Section 2.1.**

J. Bieser, A. Aulinger, V. Matthias, M. Quante, H.A.C. Denier van der Gon: Vertical emission profiles for Europe based on plume rise calculations, Environmental Pollution, Volume 159, Issue 10, 2011, Pages 2935-2946, ISSN 0269-7491, https://doi.org/10.1016/j.envpol.2011.04.030.

Fig. 4: Please add the period in the caption. Is it a summer month?

**The period has been added in the caption: this is the month of January 2015, consistently with Figure 3. Please note that Figure 4 has been updated: we now only show the absolute and relative differences between the averages of CO concentrations simulated using the prior TNO-GHGco-v3 anthropogenic emission estimate and those simulated with null CO emissions**.

difference between pirior and CO emis = 0 is very small?! Just few %.

**Yes, indeed.As mentioned above, it bears evidence that the MOPITTAK and prior profiles have a strong impact on the variations of the CO concentrations at least over Central, Eastern and Northern Europe, as mentioned above.**

l 126: "MOPITT instrument version 8". Please change: the version refers to the CO retrieval code, not the instrument.

**We have changed the sentence: « CO inversions assimilate CO observations from the MOPITT CO surface retrieval product version 8 (Deeter et al., 2019). »**

l 133: Why "surface" product instead of surface product?

**We have now clarified that the surface designation for the MOPITT product corresponds to the mean volume mixing ratio between the surface and 900 hPa, which is not exactly what the reader could expect from the label "surface" (however, since we now give the explanation, the "" have been removed). See Section 2.2: « We choose to assimilate the MOPITT V8J surface product, derived as the mean volume mixing ratio between the surface and 900 hPa... ».**

Please include a note on the sensitivity profiles or averaging kernels of MOPITT. How many degrees of signal are there typically in the combined profile retrieval?

**Buchholz et al. (2017) already present some typical averaging kernel for different pressure levels. The average degrees of freedom for signal (DFS) for the MOPITT surface product over 2004-2008 are often higher than 0.4 over Europe in autumn for the combined TIR+NIR MOPITT retrieval while they are always lower than 0.4 for the TIR only retrieval according to Worden et al., (2010) (their Figure 3).**

**We now refer to these references in Section 2.2:** « Among the different MOPITTv8 products, we choose to work with the multispectral MOPITTv8-NIR-TIR one (also called MOPITT-v8J),as the sensitivity to CO in the lower troposphere should be significantly greater for retrievals exploiting simultaneous TIR and NIR measurements than for retrievals based on either spectral region alone **(Worden et al., 2010; Deeter et al., 2013, Buckholz et al., 2017). »**

Worden, H. M., Deeter, M. N., Edwards, D. P., Gille, J. C., Drummond, J. R., and Nédélec, P. (2010), Observations of near-surface carbon monoxide from space using MOPITT multispectral retrievals, *J. Geophys. Res.*, 115, D18314, doi:10.1029/2010JD014242.

Buchholz, R. R., Deeter, M. N., Worden, H. M., Gille, J., Edwards, D. P., Hannigan, J. W., Jones, N. B., Paton-Walsh, C., Griffith, D. W. T., Smale, D., Robinson, J., Strong, K., Conway, S., Sussmann, R., Hase, F., Blumenstock, T., Mahieu, E., and Langerock, B.: Validation of MOPITT carbon monoxide using ground-based Fourier transform infrared spectrometer data from NDACC, Atmos. Meas. Tech., 10, 1927–1956, https://doi.org/10.5194/amt-10-1927-2017, 2017.

l 133: "We choose to assimilate the MOPITT "surface" product." Why?

**As explained in the introduction, Konovalov et al. (2016) derived estimates of the CO European emissions using the IASI satellite measurements and pointed out the low sensitivity of the corresponding CO total columns to the anthropogenic CO emissions over Europe. We assumed that a surface product such as that of MOPITT would be much more sensitive to these emissions and that the use of the MOPITT surface product would thus help overcome the issue documented by Konovalov et al. (2016).**

**Moreover, above mentioned studies have shown that the surface level multispectral retrievals have greater sensitivity to CO near the surface and reduced sensitivity in the free troposphere (Jiang et al ., 2015, Qu et al., 2022). The CO concentrations over 2001–2015 from the MOPITT surface product and the World Data Center for Greenhouse Gases (WDCGG) ground sites over Europe (Jiang et al., 2017) show consistent long-term trends.**

**The retrieval bias drift is also low at the surface level for V8 TIR–NIR products, as compared to National Oceanic and Atmospheric Administration (NOAA) flask measurements (Deeter et al., 2019). Finally, the surface level of the V8 TIR–NIR products gives the lowest bias when compared to situ data from NOAA aircraft validation sites (Deeter et al., 2019). We have added these sentences in the text in Section 2.2.**

Jiang, Z., Jones, D. B. A., Worden, J., Worden, H. M., Henze, D. K., and Wang, Y. X.: Regional data assimilation of multi-spectral MOPITT observations of CO over North America, Atmos. Chem. Phys., 15, 6801–6814, https://doi.org/10.5194/acp-15-6801-2015, 2015a.

Jiang, Z., Worden, J. R., Worden, H., Deeter, M., Jones, D. B. A., Arellano, A. F., and Henze, D. K.: A 15-year record of CO emissions constrained by MOPITT CO observations, Atmos. Chem. Phys., 17, 4565–4583, https://doi.org/10.5194/acp-17-4565-2017, 2017.

Qu, Z., Henze, D. K., Worden, H. M., Jiang, Z., Gaubert, B., Theys, N., & Wang, W. (2022). Sector-based top-down estimates of $NO_x$, $SO_2$, and CO emissions in East Asia. Geophysical Research Letters, 49, e2021GL096009. https://doi. org/10.1029/2021GL096009.

Would column or profile assimilation give different results?

**Jiang et al. (2015a) have found that the differences between the source estimates inferred from the profile and surface products for 2004–2005 with global inversions could be important in southern Asia, North America, and Europe. After this work, the studies of Jiang et al. (2015b) over North America and Jiang et al. (2017) over the globe only used the MOPITT surface-level. Qu et al. (2022) also assimilate the MOPITT surface-level.**

**In addition, Konovalov et al. (2016) pointed out the low sensitivity of the CO total columns (from IASI) to the anthropogenic CO emissions over Europe. The vertical sensitivity of MOPITT and IASI total columns are different, but this laysthe basis for the assumption that the inversions using MOPITT total columns would have led to corrections to the CO prior anthropogenic emissions smaller than those obtained with the surface product.**

Jiang, Z., Jones, D. B. A., Worden, H. M., and Henze, D. K.: Sensitivity of top-down CO source estimates to the modeled vertical structure in atmospheric CO, Atmos. Chem. Phys., 15, 1521–1537, https://doi.org/10.5194/acp-15-1521-2015, 2015b.

The combination of NIR and TIR holds the promise of some vertical profile information, which could be extracted when the profiles are assimilated.

**We take this information into account. Nevertheless, as we are targeting regional surface emissions, the surface product seem to be the most appropriate for our study.**

l 139: Superobservations are constructed by using the median observation in each 0.5x0.5 grid cell. I assume the median retrieval and median averaging kernel are used in the observation operator. But the process of using the median may/will remove noise from the MOPITT observations, and could result in smaller observation errors. Is the median observation error used, or a reduced error? Please state this explicitly.

**When defining such super-observations as the observation whose value is the median of the ensemble of observation within a 0.5°x0.5° grid cell of the CTM, and within the CTM**

**physical time steps, we assign the observation error associated to this observation to the super-observation, which is implicitly a conservative observation error derivation. We assume that a large part of the errors on the individual retrievals are correlated between the different retrievals over short spatial scales (i.e. that a large part of the error consists in "systematic errors" due to local biases in the retrievals).**

**In any case, the typical number of MOPITT observations per CTM grid cell is often about 1 or 2or 3 observations. Cases when it is 4 to 6 are relatively rare, so that the choice to decrease the error on the super-observation as a function of the number of observation or not do not imply large changes in most of the cases.**

**We have changed the sentences in Section 2.2: « In order to associate the super-observations to a real AK, the super-observations have been taken as the individual observation corresponding to the value of the median of the MOPITT concentrations within the 0.5°x0.5°grid-cell of the CTM and within the CTM physical time steps (about 5-10 min). The AK and the uncertainty associated to this individual value are then used to define the AK and uncertainty for the « super-observation ». In principle, the observation error associated to such a median value should be smaller than the error associated to individual observation, but, here, we keep the error for the individual observation used to define the super-observation as a conservative estimate of the super-observation error. The super-observations therefore do not have a smaller error than the individual observations as mentioned above»**

l 140: What is "AK" (not defined before this point)? Please discuss the averaging kernels: are these used in the observation operator or not? I assume they are.

**The averaging kernels (AKs) are an indication of the vertical sensitivity of the measurements and of the amplitude of the corrections applied to the retrieval prior vertical profile to derive CO observations. They are included in the observation operator.**
**We have added information in the text: « To make accurate comparisons between simulations and satellite observations the averaging kernels (AKs) and the MOPITT prior profiles are applied to the simulated field so that the resulting simulation of the concentrations exhibit the same degree of smoothing and a priori dependence as the MOPITT product (Deeter et al., 2013; Deeter et al., 2019).**
**Following the recommendations of Deeter (2018), the formula is applied:**
**$c_m = x_a + AK(c_m° - x_a)$ (Eq1)**
**where:**
**– $c_m$ is the modeled column,**

**– AK contains the averaging kernels -which are an indication of the vertical resolution of the measurements- provided in the form of a matrix,**

**– $x_a$ is the prior profile derived from a model climatology and vary seasonally and geographically (Deeter et al., 2019)**

**– and $c_m°$is the vertical distribution of the original model partial columns interpolated to the pressure grid of the AKs. »**

I would find it useful if some typical AK profiles are shown, in order to better understand Fig.3. Adding the modelled surface concentration and column to Fig.3 would be helpful.

**Some typical AK profiles are shown in Buchholz et al. (2017) and this reference has been added in the text in Section 2.2, as mentioned above. Here, we have added the modelled « surface » (surface-to-900hPa) concentrations from CHIMERE without applying the MOPITT AK and prior profiles in Figure 3. It is interesting as it shows the impact of the retrieval formula (Eq1 as mentioned above) on the simulated concentrations, including the weighting of vertical integration by AKs.**

Sec 2.2. How is the uncertainty of the superobservation determined. Is it equal to the error of the median retrieval? A superobservation may have a smaller error than the individual observations. Since the uncertainty is important for the final result, the authors should explain this more clearly.

**As recalled above, the super-observations have been taken as the individual observation corresponding to the value of the median of the MOPITT concentrations within the 0.5°x0.5°grid-cell of the CTM and within the CTM physical time steps (about 5-10 min). The uncertainty associated to this individual value becomes the uncertainty of the super-observation. As mentioned above, in our case, the super-observations therefore do not have a smaller error than the individual observations as mentioned above.**

l 154: Why are emissions specified for 8 levels? Is this needed?

**The 8 levels are needed to represent the range of injection heights for the largest point sources (e.g., mainly for the energy production and industrial sector) provided in the TNO emission inventory, as accounting for elevated emissions may be critical, as shown in Brunner et al. (2019) for $CO_2$.**

**This information has been added in the text in Section 2.3:** «CO anthropogenic emissions at a 1-day temporal resolution, at a 0.5°×0.5° resolution andover the first 8 vertical levels of CHIMERE, i.e., for a one-month inversion, for each of thecorresponding (28 to 31 days)×101×85×8 grid cells. **The 8 levels are needed to represent the range of injection heights for the largest point sources, (e.g., mainly for the energy production and industrial sector) provided in the TNO emission inventory.**»

Brunner, D., Kuhlmann, G., Marshall, J., Clément, V., Fuhrer, O., Broquet, G., Löscher, A., and Meijer, Y.: Accounting for the vertical distribution of emissions in atmospheric $CO_2$ simulations, Atmos. Chem. Phys., 19, 4541–4559, https://doi.org/10.5194/acp-19-4541-2019, 2019.

Does the analysis significantly change the vertical distribution of emissions?

**No, the inversions do not significantly change the vertical distribution of the emissions. Please note that the CO emissions shown in the paper corresponds to the sum of the emissions on different levels.**

Sec 2.3: The consistency between B, R and the observation-minus forecast departures can be tested using chi^2. From the paper I have the impression this was not done. Why not?

The χ2 value is used to diagnose balance between actual errors and estimated errors. Our χ2 is of about 0.2 for January 2015.When the χ2 value is smaller than the ideal value of 1, it suggests overestimated background error covariance or observation errors. This χ2may therefore confirm that our set-up of the uncertainties in the covariance matrices R are too large (see the discussion above regarding the assignment of the observation error for super-observations). However, the typical factor of decrease of the observation error for super-observation when accounting for the noise component of the individual retrieval errors may hardly explain that the χ2 is about 0.2. Actually, we also have an important level of prior uncertainty at the model grid cell and 1-day scale (100%), i.e. the balance between the prior and observation uncertainties is not necessarily misrepresented. The robustness of the interpretation of the χ2 diagnostic can thus hardly be used to correct the observation errors, for which we have tried to assign a sensible level based on the reported retrieval errors.

We now indicate in the conclusion that our derivation of the error associated with each super-observation could be conservative as mentioned above, and we discuss the potential and challenges associated to the improvement of the assignment of observation errors for individual observations (Gaubert et al., 2023) and for super-observations.

Gaubert, B.; Edwards, D.P.; Anderson, J.L.; Arellano, A.F.; Barré, J.;  Buchholz, R.R.; Darras, S.; Emmons, L.K.; Fillmore, D.; Granier, C.; et al. Global Scale Inversions from MOPITT CO and MODIS AOD. Remote Sens.2023, 15, 4813. https://doi.org/10.3390/rs15194813

Sec 3: I was surprised to read that emissions of CO need to be reduced. In the past models have struggled with a low bias in CO, e.g. Stein, 10.5194/acp-14-9295-2014, suggesting emissions should be increased, especially in winter. In Fig. 3 Chimere is generally higher than MOPITT.

Global models have indeed struggled with a low bias in CO in the Northern Hemisphere, particularly in winter (Fortems-Cheiney et al., 2011, Stein et al., 2014), suggesting that emissions should be increased by the inversion.  Compared to these previous estimates, we have increased the spatial resolution of our transport model, used improved emission inventories as prior estimates of the inversions, and assimilated the recent MOPITT v8 observations. Therefore, the context for the correction of the inventories in our regional inversions is totally different from the past global analysis.
The discrepancies between simulated and observed CO concentrations can indeed be explained by the inventory used for the estimation of the prior CO emissions. Stein et al. (2014) have shown that the quantification of the emissions can be very different depending on the bottom-up inventories (e.g, 118 vs 70 TgCO respectively from ACCMIP and for RETRO for North America in year 2000, their Table 1). In addition, updated inventories were not always available at the time of different studies and it could explain the need to increase the CO emissions by the inversions, particularly when the gap between the studied year and the reference year of the inventory was large in the 2000s and in the 2010s. Here, we used the TNO-GHGco-v3 inventory based on a recent EMEP/CEIP official country reporting for air pollutants, specifically covering the years we are studying, with the exception of years 2020 and 2021.Model errors in long-range transport, diffusion, chemistry (linked to the radical hydroxil OH and to NMVOCs) and coarse resolution (Valin et al., 2011) can also all impact the inverse modeling of CO at the global scale.

**We have added the following sentences in the text in Section 3.1: « It is interesting to note that global models have struggled with a low bias in CO in the Northern Hemisphere, particularly in winter, compared to the MOPITT observations (Fortems-Cheiney et al., 2011; Stein et al., 2014), while our inversions results tend to decrease the emissions compared to the inventory we use as a prior estimate for the inversion. However, compared to these previous studies, we have used more recent MOPITT observations and validation results for version 8 MOPITT CO products indicate reduced long-term bias drift, weaker bias geographical variability and smaller biases overall compared to version 7 (Deeter et al., 2018).We have also used a more recent TNO-GHGco-v3 CO emission inventories as prior estimates for the inversion, this inventory being based on one of the latest EMEP/CEIP official country reporting for air pollutants. As model errors in long-range transport, diffusion, chemistry linked to the radical hydroxil OH and to NMVOCs (Strode et al., 2015) and coarse resolution (Valin et al., 2011) can all impact the inverse modeling of CO (Arellano Jr. et al., 2006; Fortems-Cheiney et al., 2011; Jiang et al., 2017; Zheng et al., 2019), we also used a chemical scheme describing the CO chemistry (including its secondary production through the oxidation andphotolysis of hydrocarbons and its sink with OH, Section 2.1) and we have increased thespatial resolution of the transport model with a regional CTM. These different aspects can explain that our regional inversion do not highlight a low bias in the inventories, unlike past global inversions studies. »**

Arellano, A. F.Jr., P. S. Kasibhatla, L. Giglio, G. R. van der Werf, J. T. Randerson, and G. J. Collatz (2006), Time-dependent inversion estimates of global biomass-burning CO emissions using Measurement of Pollution in the Troposphere (MOPITT) measurements, *J. Geophys. Res.*, 111, D09303, doi:10.1029/2005JD006613.

Fortems-Cheiney, A., F. Chevallier, I. Pison, P. Bousquet, S. Szopa, M. N. Deeter, and C. Clerbaux (2011), Ten years of CO emissions as seen from Measurements of Pollution in the Troposphere (MOPITT), *J. Geophys. Res.*, 116, D05304, doi:10.1029/2010JD014416.

Gaubert, B., Emmons, L. K., Raeder, K., Tilmes, S., Miyazaki, K., Arellano Jr., A. F., Elguindi, N., Granier, C., Tang, W., Barré, J., Worden, H. M., Buchholz, R. R., Edwards, D. P., Franke, P., Anderson, J. L., Saunois, M., Schroeder, J., Woo, J.-H., Simpson, I. J., Blake, D. R., Meinardi, S., Wennberg, P. O., Crounse, J., Teng, A., Kim, M., Dickerson, R. R., He, H., Ren, X., Pusede, S. E., and Diskin, G. S.: Correcting model biases of CO in East Asia: impact on oxidant distributions during KORUS-AQ, Atmos. Chem. Phys., 20, 14617–14647, https://doi.org/10.5194/acp-20-14617-2020, 2020.

Strode, S. A., Duncan, B. N., Yegorova, E. A., Kouatchou, J., Ziemke, J. R., and Douglass, A. R.: Implications of carbon monoxide bias for methane lifetime and atmospheric composition in chemistry climate models, Atmos. Chem. Phys., 15, 11789–11805, https://doi.org/10.5194/acp-15-11789-2015, 2015.

Valin, L. C., Russell, A. R., Hudman, R. C., and Cohen, R. C.: Effects of model resolution on the interpretation of satellite NO2 observations, Atmos. Chem. Phys., 11, 11647–11655, https://doi.org/10.5194/acp-11-11647-2011, 2011.

Zheng, B., Chevallier, F., Yin, Y., Ciais, P., Fortems-Cheiney, A., Deeter, M. N., Parker, R. J., Wang, Y., Worden, H. M., and Zhao, Y.: Global atmospheric carbon monoxide budget 2000–2017 inferred from multi-species atmospheric inversions, Earth Syst. Sci. Data, 11, 1411–1436, https://doi.org/10.5194/essd-11-1411-2019, 2019.

But in the 2021 paper, fig.5, Chimere simulates smaller concentrations. Please explain the increase in CO in the model. Is it linked to the emissions, boundaries or model change?

**There is no systematic bias in the seasonal cycle of bottom-up inventory, so the results for one week in March can be different to the results of one month in January 2015. In addition, improvements have been made since the study of Fortems-Cheiney (2021), including the use of a more recent inventory (TNO-GHGco-v3 instead of TNO-GHGco-v1). We have added this information in Section 2.1: « Contrarily to Fortems-Cheiney et al. (2021) using the TNO-GHGco-v1, the prior estimate of CO anthropogenic emissions is derived from the recent TNO-GHGco-v3 gridded inventory for the period 2011-2018. »**

Please add some discussion on modelling bias in CO, including a few relevant papers.

**We now stated, as mentioned above, that model errors in long-range transport, diffusion, chemistry (e.g., linked to the radical hydroxil OH and to NMVOCs, Strode et al., (2015)) and coarse resolution (Valin et al., 2011) can all impact the inverse modeling of CO (Arellano Jr. et al., 2006; Fortems-Cheiney et al., 2011; Jiang et al., 2017 ; Zheng et al., 2019).**

Sec 3: Fortems-Cheiney (2021) mentions "local increments on CO emissions can reach more than +50 %, with increases located mainly over central and eastern Europe, except in the south of Poland, and decreases located over Spain and Portugal." Please contrast these earlier results with the results presented here.

**The illustration in Fortems-Cheiney (2021) presents results for another period, with different prior inventory and prior error statistics. This can explain the differences in the corrections provided by the inversions. We have added information in Section 2.3: «Contrarily to Fortems-Cheiney et al. (2021) where they are set to 15%, the ratios between the prior error standard deviations in B and the prior estimates are set at 50% for theCO lateral conditions. »**

Fig 3. In data assimilation the prior for a given month is often based on the posterior from the previous month. In contrast, the prior could also refer to a free model run with prior emissions. Maybe I missed it, but it was not clear to me how "prior" is defined in this paper. Please clarify if the results (e.g. emission adjustment) is passed on from one month to the next in the assimilation.

**Our 1-month inversions are independent of each other: the initial conditions for a given month are not based on the posterior from the previous month. This has been added in the text: « As a trade off between computational resources and relevance of our inversions with a moderate impact of the initial conditions on our 1-month CO simulation, series of independent 1-month inversion windows are run. We therefore do not account for the**

**potential update of the concentrations during a previous 1-month window due to the inversions. »**

**Corrections on emissions during a given month are not projected onto emissions for the following month, because there is no strong reason to believe that the error on emissions is strongly correlated from one month to another.**

l 233: "The posterior CO emissions display a very similar decreasing trend than the prior emissions over the EU-27+UK area" If the assimilation uses the trends in the prior TNO emission database, and if the observations do not provide a strong forcing, then this may not be very surprising. (Would be nice to have results using fixed emissions, e.g. using 2015.)

**The posterior CO emissions indeed display a very similar decreasing trend than the prior emissions over the EU-27+UK area and it is explained by the lack of large-scale corrections. Nevertheless, as described in Section 3.2, differences are found in areas benefiting from the best MOPITT coverage. Consequently, the assimilation of MOPITT observations in the inversions can attenuate the strong decreasing trend of the CO emissions in the TNO-GHGco-v3 inventory, particularly during autumn and winter, over areas benefiting from the best MOPITT coverage.**

**It would indeed be nice to performed inversions with a constant prior emissions over the 10-year period but it would have been computationally expensive.**

l 203: "posterior simulation still presents positive biases" In fact only a minor part of the bias (order 20% over the continent) is removed if I understand Fig. 3-d correctly. Please be more quantitative here and mention these percentages.

**The percentage has been added: « The mean bias over the entire domain between the simulation and the MOPITT super-observations is reduced by about 2%. Nevertheless, the corrections made to the prior TNO-GHGco-v3 inventory are particularly large in areas where both CO emissions and the sensitivity of CO concentrations to the emissions are high. For example, the posterior emissions reduce the mean bias between simulated concentrations and MOPITT data by about 26% over the Po Valley in Italy and over Benelux in January 2015 (Figure 3d). »**

l 213: Figure 1c should be 2b, I assume?

**Indeed. We have changed the sentence:  « The highest increments are found over large cities and over industrial areas (Figure 2b), where the CO emissions are high (Figure 2a).**

l 216: "Table 3" should be Table 2 I assume.

**Yes indeed. This has been corrected.**

l 239: "steadily increasing" Please provide trend figures per country for 2011-2019 (extra table or combined in Table 2).

**We have changed the sentence:« While the TNO-GHGco-v3 inventory shows significant decreasing trends in these regions, the posterior emissions appear to be stagnating, with even non significant increasing trend over parts of Italy. »**

**We think that Figure 6 is already informative. We choose not to provide trend figures per country.**

l 262: "decrease by about -1.3%" Could this be related to the remark on lime 203 that posterior emissions are still biased and capture only part of the model-MOPITT difference?

**The decrease by about 1-3% is related to the EU-27+UK area and is indeed partly explained by the fact that the posterior emissions capture only a part of the model-MOPITT difference. Nevertheless, the inversions lead to a higher decrease of CO emissions over areas where the anthropogenic emissions are usually large, and particularly over industrial basins such as over the Benelux, over the Rhine-Rhur Valley and over the Po Valley.**

Fig.7. Why is the sea/ocean yellow in 2020 (positive increment) and not in 2019?

**This was due to a land-sea mask applied to one of the subfigure only. This has been corrected.**

---

## Author Response (AR2)

**Reviewer #1**

**We wish to thank the referee for his/her helpful comments. His/her full comments are copied hereafter in normal black font, and our responses are inserted in between in bold font.**

Here are some minor comments:

P10L197: "The potential of MOPITT to provide information can also be hampered by the errors associated to the MOPITT super-observations (Figure 5b, see Section 2.2)." The sentence itself is in Section 2.2, please remove.

**It has been done.**

Also here, can you mention that MOPITT V9J has a better observation coverage in polluted areas.

Deeter, M., Francis, G., Gille, J., Mao, D., Martínez-Alonso, S., Worden, H., Ziskin, D., Drummond, J., Commane, R., Diskin, G., and McKain, K.: The MOPITT Version 9 CO product: sampling enhancements and validation, Atmos. Meas. Tech., 15, 2325–2344, https://doi.org/10.5194/amt-15-2325-2022, 2022.

**It has been done. We have added the following sentence: « The new MOPITT retrieval Version 9 product has a better observation coverage, with a number of daytime MOPITT retrievals over land increased by 30%–40% relative to the Version 8 product due to improvments of the cloud detection algorithm (Deeter et al., 2022). »**

Line 271: Change "radical hydroxil" to hydroxyl radical.

**It has been done.**

Line 289: Why is "super-observations" bold ?

**It was an oversight of the track-changes version. It has been corrected.**